# Immunogenetic losses co-occurred with seahorse male pregnancy and mutation in *tlx1* accompanied functional asplenia

Yali Liu[1,2,3,8], Meng Qu[1,2,8], Han Jiang[1,3,8], Ralf Schneider [4,8], Geng Qin [1,2], Wei Luo [1], Haiyan Yu[1], Bo Zhang[1], Xin Wang[1,2], Yanhong Zhang[1,2], Huixian Zhang[1,2], Zhixin Zhang[1,5], Yongli Wu[1], Yingyi Zhang[1,3], Jianping Yin[1,2], Si Zhang[1,2], Byrappa Venkatesh [6], Olivia Roth [4] ✉, Axel Meyer [7] ✉ & Qiang Lin [1,2,3] ✉

In the highly derived syngnathid fishes (pipefishes, seadragons & seahorses), the evolution of sex-role reversed brooding behavior culminated in the seahorse lineage's male pregnancy, whose males feature a specialized brood pouch into which females deposit eggs during mating. Then, eggs are intimately engulfed by a placenta-like tissue that facilitates gas and nutrient exchange. As fathers immunologically tolerate allogenic embryos, it was suggested that male pregnancy co-evolved with specific immunological adaptations. Indeed, here we show that a specific amino-acid replacement in the *tlx1* transcription factor is associated with seahorses' asplenia (loss of spleen, an organ central in the immune system), as confirmed by a CRISPR-Cas9 experiment using zebrafish. Comparative genomics across the syngnathid phylogeny revealed that the complexity of the immune system gene repertoire decreases as parental care intensity increases. The synchronous evolution of immunogenetic alterations and male pregnancy supports the notion that male pregnancy co-evolved with the immunological tolerance of the embryo.

The evolutionary diversification of animals went hand-in-hand with an increasing complexity of the immune system[1,2]. As a hallmark of vertebrate evolution, the MHC/B-cell receptor/T-cell receptor system, an essential arm of the adaptive immune system, first appeared in jawed vertebrates and accompanied both the radiation of sharks and rays, and later the radiation of bony fishes[3,4]. The emergence of specialised cells and molecules, as well as the lymphatic system and the spleen as an important vertebrate secondary lymphoid organ permitted complex immunological reorganisation and modifications during the evolution of the vertebrate adaptive immune system[2]. As a rare exception, the spleen is absent in seahorses (Family Syngnathidae)[5,6]. This raises the question of how they cope without this vital immune-organ and what ultimately selected for the evolutionary loss of the spleen. Seahorses are famous for their iconic morphology and their highly unusual life history, which includes sex-role reversed brooding behavior via "male pregnancy"[7,8]. While females in more basal lineages of syngnathids simply glue their eggs to brooding patches on the ventral side of males, the males' seahorse brood pouch represents a

[1]CAS Key Laboratory of Tropical Marine Bio-Resources and Ecology, South China Sea Institute of Oceanology, Chinese Academy of Sciences, 510301 Guangzhou, China. [2]Guangdong Provincial Key Laboratory of Applied Marine Biology, South China Sea Institute of Oceanology, Chinese Academy of Sciences, Guangzhou 510301, PR China. [3]University of Chinese Academy of Sciences, 100101 Beijing, China. [4]Marine Evolutionary Ecology, Zoological Institute, Kiel University, 24118 Kiel, Germany. [5]Graduate School of Marine Science and Technology, Tokyo University of Marine Science and Technology, Minato, Tokyo, Japan. [6]Institute of Molecular and Cell Biology, A*STAR, 138673 Singapore, Singapore. [7]Department of Biology, University of Konstanz, 78464 Konstanz, Germany. [8]These authors contributed equally: Yali Liu, Meng Qu, Han Jiang, Ralf Schneider. ✉e-mail: oroth@zoologie.uni-kiel.de; axel.meyer@uni-konstanz.de; linqiang@scsio.ac.cn

more derived organ for paternal care and is the most complex structure in their family to protect and nourish embryos[9]. Female seahorses transfer eggs during mating into the males' brood pouches where embryos are implanted and nourished by a "pseudoplacenta". Its function is analogous to a mammalian maternal placenta and provides nutrients and oxygen to the developing embryos that hatch inside the males' pouch[10–12] (Fig. 1a).

In vertebrates, viviparity in females—with the unique exception of the sex-role reversed seahorses' male pregnancy—has evolved over 150 times independently[13,14]. While pregnancy provides advantages to the developing offspring, allowing them to be better protected from early-life predation and to be released at an advanced life-history stage, it poses an immunological challenge for the pregnant parent: How are the semi-allogenic

embryos immunologically tolerated? As a solution to this immunological challenge mammalian embryos reduce the diversity of MHCI molecules expressed on trophoblasts, which constitute the cell layer in direct contact with maternal tissue[4,15]. In contrast, in seahorses, the co-evolution of the immune system with male pregnancy remains largely unknown. In addition to the spleen, some other important parts of the adaptive immune system's genetic repertoire are absent in seahorses, and these secondary losses have been hypothesized to be linked to the evolutionary novelty that is male pregnancy[5,6].

In an effort to study the immune-related changes during the evolution of seahorse male pregnancy, we comparatively analyze the genomes of two de novo sequenced seahorses together with other teleost genomes that had been previously published, focusing on the

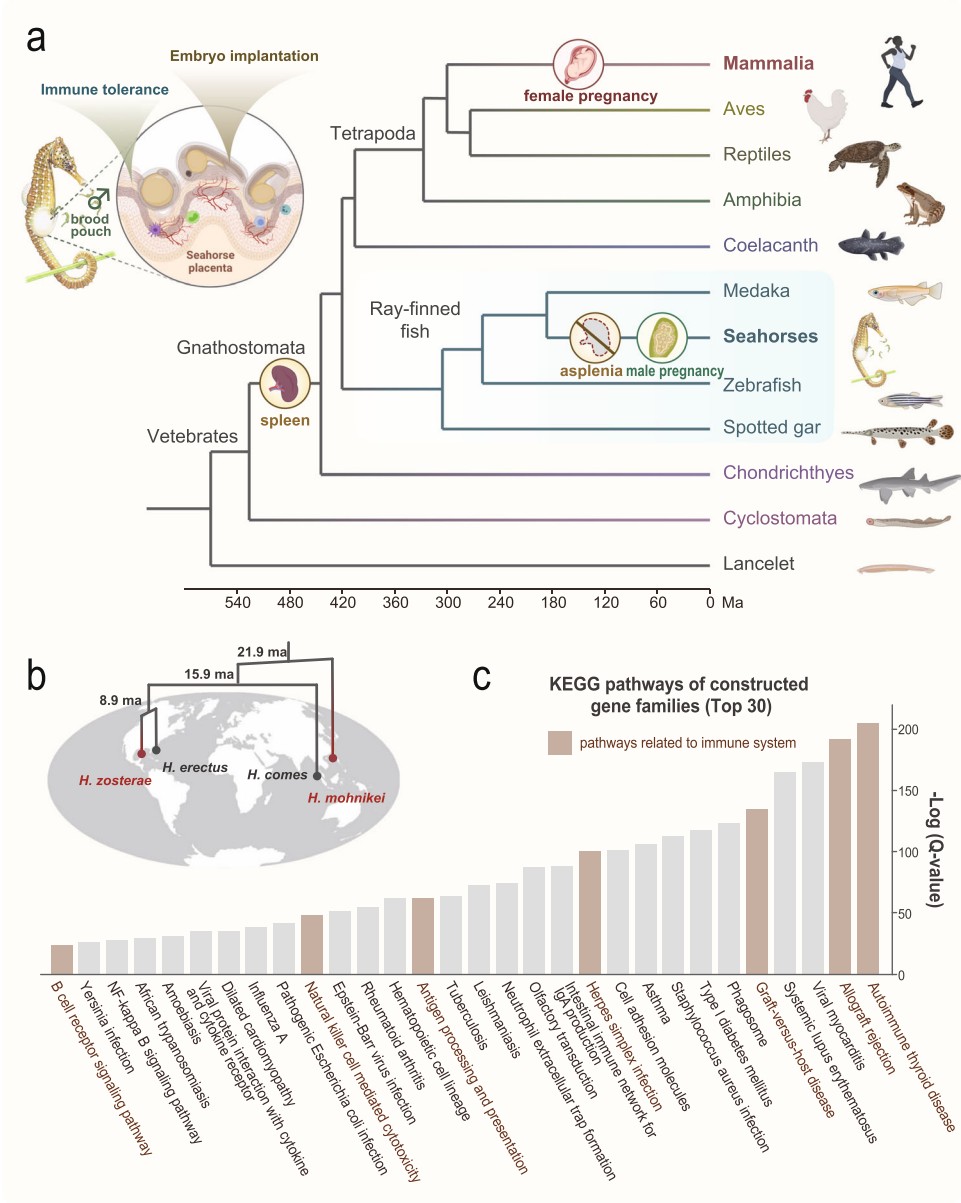

**Fig. 1 | Unique features of immunity and reproduction in seahorses. a** A species tree showing the evolution of specialized male pregnancy and asplenia traits in the seahorse. During pregnancy, embryos implanted in the "pseudoplacenta" of male seahorses are recognized by the paternal immune system. **b** Four seahorse species, from the major lineages of seahorses were included in the comparative genomic analyses. Newly sequenced species are indicated in red. Ma, million years ago. **c** The

top 30 KEGG pathways of contracted gene families in seahorses. Categories involved in immunity are colored in brown. The figures were created with BioRender.com. The used map was downloaded from a free world map website (https://www.freeworldmaps.net/outline/maps.html). Source data are provided as a Source Data file.

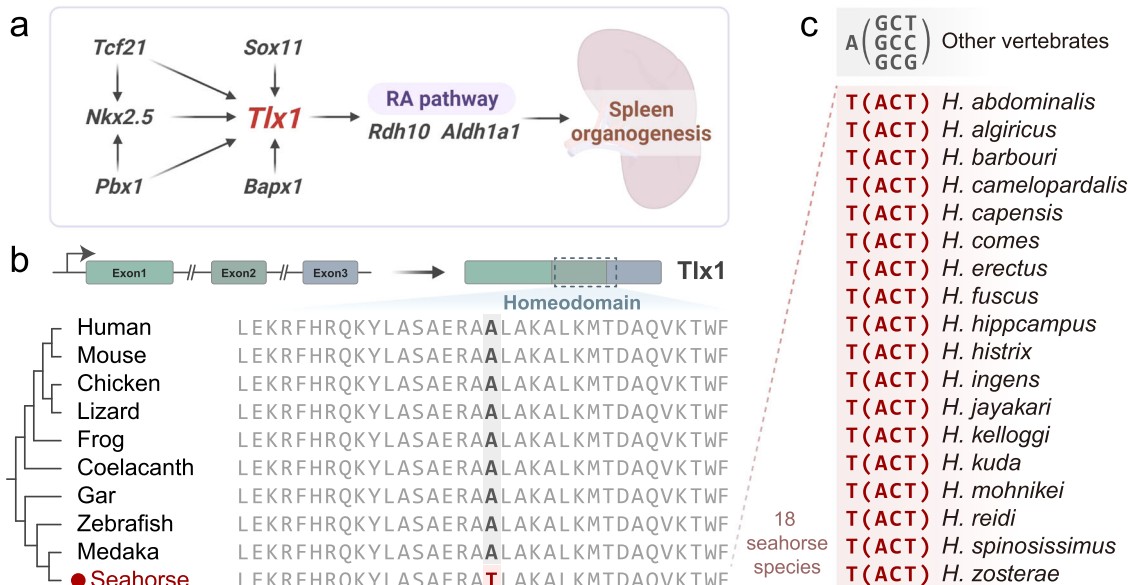

**Fig. 2 | Seahorse-specific mutation of the *tlx1* gene. a** Schematic diagram of the involved genes during spleen growth and morphogenesis (modified from previous studies[18]). Red color Tlx1, the sequence-specific variation in seahorses; RA, retinoic acid. **b** Gene structure (upper panel) and multiple amino acid alignments (lower panel) of Tlx1. Seahorse harbors a missense mutation of A (Alanine, Ala) to T (Threonine, Thr) in the homeodomain. **c** Extended Data examination of the identified *tlx1* mutation in 18 seahorse species. All detected seahorses exhibit T (ACT) instead of A (GCT or GCC or GCG), unlike other vertebrates. The figure was created with BioRender.com.

evolutionary innovation of the immune-related genes (Fig. 1b and Supplementary Table 1). We discover that a single amino acid mutation in the *tlx1* transcription factor is most likely causal for the most drastic change, the loss of the spleen (termed "asplenia") in seahorses. This we verify by knockout and missense mutations of *tlx1* in zebrafish. Novel insights into the modified immune system, including the complement system and immunoglobulins in this lineage support its previously hypothesized link to the evolutionary origin of male pregnancy.

## Results

### Contraction and loss of immune-related gene families in seahorse genomes

We generated chromosome-level reference genomes of *Hippocampus zosterae* (2n = 44) and *H. mohnikei* spanning 458 Mb and 527 Mb, respectively. With contig N50 of 7.3 Mb and 18.1 Mb and nearly complete BUSCO genes (92.95% and 93.46%), we succeeded in generating high-quality assemblies (Supplementary Table 5, 6, Supplementary Note 1). By including the already published genomes of *H. comes*[16] and *H. erectus*[17], we could comprehensively conduct comparative genomics analyses of four seahorse species (subfamily Syngnathinae) that evolved male pregnancy (Fig. 1b) by comparison with nine other teleost genomes from different orders (Supplementary Note 2). We show that 1260 gene families have undergone significant contraction in the seahorse lineage (Supplementary Fig. 4a and Supplementary Data 1–2). KEGG pathway enrichment analysis of the contracted gene families revealed signatures related to immune response pathways, such as the autoimmune thyroid disease, allograft rejection, and antigen processing and presentation (Fig. 1c and Supplementary Table 13), which was supported by a gene loss analysis (Supplementary Fig. 4b and Supplementary Data 1). The contraction or loss of immune-related gene families have substantially contributed to the modifications identified in the seahorse immune system and are presumably linked to the unique evolution of male pregnancy and its brood pouch structure (Supplementary Note 3). As to none of these genes, a role in spleen development can be assigned, their loss cannot be the molecular basis for asplenia in seahorses.

### Lineage-specific missense mutation of *tlx1* in *Hippocampus* and *Syngnathus*

Our comparative genomic analysis identified genes undergoing positive selection, rapid evolution, or containing lineage-specific mutations in seahorses (Supplementary Data 4–8). Spleen development is mediated by the precise regulation of several transcription factors, including *Pbx1*, *Tlx1*, *Nkx3-2*, and *Nkx2-5*[18–20] (Fig. 2a). In the homeobox domain of *tlx1* (initially mistaken for *Hox11*[19,21]) a seahorse-specific mutation was identified (Fig. 2a and Supplementary Figs. 8–10). *Tlx1*, which contains three exons and a homeobox domain (Fig. 2b), has been reported to control splenic primordia cell fate specification and organ expansion. It is the only known gene whose pseudofunctionalization results in spleen loss without causing other developmental abnormalities[19]. To validate the specificity of the mutation, we extended our analyses to a larger phylogenetic range (34 species), including mammals, birds, reptiles, amphibians, and additional fishes. The amino acid sequences of *tlx1's* homeobox were identical in all vertebrates except for a seahorse-specific mutation, where a hydrophilic threonine (T, Thr) has been replaced by the hydrophobic alanine (A, Ala) (Fig. 2b and Supplementary Fig. 11a). Further validation of *tlx1* exon2 in 18 seahorse species confirmed this mutation to be a common feature of all seahorses (Fig. 2c, Supplementary Figs. 12–13 and Supplementary Note 4). The family *Syngnathidae* is a large (>350 species) and diverse clade of morphologically unique teleosts. They can be divided into two subfamilies: the *Nerophinae* and the *Syngnathinae*[22,23]. When assessing the mutation's locus also in all other available members of the subfamily Syngnathinae, we detected that all investigated members of the genus *Syngnathus* (all with quite derived, closed brooding organs and closely related to seahorses[8]) share the described mutation. In contrast, more distantly related members with less complex brooding organs, such as the seadragon and the alligator pipefish – where the eggs are simply attached to the ventral side of the males' tail[8] – have retained the ancestral Alanine in this position. As for the subfamily *Nerophinae*, we found the TLX1 sequences exhibit amino acid substitutions of A to L and A to I in *Oostethus manadensis* and *Nerophis ophidion*, respectively (Supplementary Fig. 11b). We also

provided the splenic phenotype for a number of syngnathid species using morphological, histological, and Micro CT methods. Dissections showed that species of the genera *Hippocampus* and *Syngnathus* (both belonging to Syngnathinae) have evolutionarily lost an unambiguous spleen, but not *Syngnathoides biaculeatus* (belongs to the subfamily *Syngnathinae*) nor *Nerophis ophidion* (that belong to the subfamily *Nerophinae*) (Supplementary Fig. 14). As mutations in exons of protein-coding genes can lead to substantial phenotypic changes, we hypothesized that the identified mutation in the conserved homeobox domain of *tlx1* might be associated with the loss of the spleen in the *Hippocampus* and *Syngnathus* species.

### Missense mutation of *tlx1* leads to an asplenia phenotype

To test this hypothesis, we conducted CRISPR/Cas9-mediated genome editing to generate two zebrafish lineages: the full gene knockout line (*tlx1*▲) and the specific point-mutation line (*tlx1*^A208T) mimicking the seahorse missense mutation (Fig. 3a, Supplementary Fig. 15 and Supplementary Note 5). As shown in Fig. 3b and c, both *tlx1*▲ and *tlx1*^A208T zebrafish were found to lose their spleen in all examined individuals, suggesting that the seahorse-specific mutation is causally linked to a functional alteration of *tlx1*, which leads to functional asplenia. Whether this mutation originally caused functional asplenia in this syngnathid lineage, or whether another mutation was causal, removed stabilizing selection from genes involved in spleen development, and *tlx1* mutated only then, cannot be resolved using our data-set. Previous studies in both mammals and zebrafish have shown that *tlx1* knockouts result in congenital asplenia[19,24], indicating a crucial and conserved vertebrate developmental pathway. In an additional zebrafish lineage, another A to T point mutation was produced at the site adjacent to the seahorse-specific mutation as a control point mutation. Correspondingly, the *tlx1*^A207T line had a normal spleen, similar to that in the wild-type zebrafish, lending further support to the hypothesis that the A208T mutation in seahorses might be causal for the loss of the spleen (Fig. 3b, c and Supplementary Figs. 16–18).

The transcription factor *tlx1* contributes not only to spleen organogenesis but also has a crucial role in brain development and function[25]. In situ experiments of seahorse *tlx1* showed expression in the hindbrain and pharyngeal arches in embryos, similar to that in mice and zebrafish during embryogenesis (Supplementary Fig. 19a and Supplementary Note 6)[19,21,24]. To investigate the effects of seahorse-specific mutants and compare them with the full gene knockout line, transcriptomic-wide gene expression analyses (RNAseq) of the brain and three immune-related organs (liver, kidney, and intestine) were performed (Supplementary Note 7). Compared to the *tlx1*▲ line, the *tlx1*^A208T line always shows transcriptomic patterns more similar to those of the wild type, with a lower number of differentially expressed genes (DEGs) in all tested tissues (Fig. 3d, Supplementary Fig. 19b–e and Supplementary Data 11). Compared with the wild-type zebrafish, the asplenia zebrafish (*tlx1*^A208T) exhibited differential gene expression profiles of genes involved in the MHC and complement pathways, e.g., *mhc1zka* and *ciita* downregulated, while *chia*, *C3a.2*, and *C9* were upregulated specifically in the kidney (Supplementary Fig. 19i–k).

In addition, most DEGs were specific for either *tlx1*▲ or *tlx1*^A208T, except for a small number shared by both lines (75 genes, brain), indicating that more severe effects are caused by the full gene knockout compared to the point mutation (Fig. 3e, Supplementary Fig. 19f–h and Supplementary Data 11). GO enrichment analysis of DEGs in the brain between *tlx1*▲ and *tlx1*^A208T lines revealed signatures in the membrane component, oxidase enzyme activity, and biosynthetic processes (Fig. 3f, Supplementary Fig. 20 and Supplementary Data 12), which are vital for brain development and function[26,27]. Our results suggest that although *tlx1*▲ and *tlx1*^A208T zebrafish had the same asplenia phenotype, genome editing led to different effects on other organs.

### Evolutionary immunogenomic modification of the unique male pregnancy

In an effort to test the hypothesis that regressive evolution of the immune system is linked to the evolution of unique male-pregnancy, we scanned genomes for modifications in immune-related genes involved in pregnancy informed by previous publications[15,28–32] and recorded their complete loss, reduced copy number or other noticeable sequence variation (Fig. 4a and Supplementary Note 8). In this study, we observed ubiquitous presence of MHCII molecules (three copies of MHC IIa and three to four copies of MHC IIb) and a conspicuous loss of not only *cd8b*[6], but also *batf3* in seahorses (Fig. 4a, Supplementary Figs. 21a, 22). Meanwhile, *il12a/b* and *ifng* genes, which encode the cytokines that are important for the alternative Batf3-independent pathway[33], are present in seahorses and pipefishes (Fig. 4a).

The spleen is essential for producing B-1a B lymphocytes, which are responsible for the production of natural antibodies in mammals[34]. Importantly, also *cd5*, the gene homologous to mammalian CD5, which encodes the surface molecule of B-1a B cells, was lost in seahorses (Fig. 4, Supplementary Figs. 21b, 23). In addition, we found that one of the necessary immunoreceptor tyrosine-based activation motifs (ITAMs) in *cd79a* is absent in seahorses (Fig. 4, Supplementary Figs. 24–25). As a vital element of the B cell receptor complex, the structural variation of *cd79a* might change the signaling ability of B cell receptors[35]. The hallmark of the vertebrate adaptive immune system is the somatic diversification of the immunoglobulin family (VDJ-rearrangement) as an advancement of the innate immune system[36]. Herein, we found a significantly reduced number of V domains in the heavy chain coding genes (*ighv*) in seahorses and other syngnathid fishes (only 2–20) compared to other teleosts, including tilapia, platyfish, medaka, stickleback, fugu, zebrafish and spotted gar (≥35) (Fig. 4 and Supplementary Fig. 26). In addition, antibodies can induce rejection of non-self through activation of the classical complement system, especially the complement component 4d (C4d)[37]. In mammals, several pregnancy complications are associated with excessive or misdirected activation of the complement system[38]. In this study, we found only two copies of the *C3* gene in seahorses - fewer than in all other teleosts (three to eight copies) - while *C4* is entirely absent (Fig. 4, Supplementary Figs. 21c, 28a). Finally, in mammals, immune tolerance is enhanced by increasing Treg cells during pregnancy[39]. Our study revealed that *foxp3* - a crucial regulator in the establishment and maintenance of Treg phenotypes[40] - was also lost in seahorses (Fig. 4 and Supplementary Fig. 28b). Moreover, several vital genes involved in pregnancy including *cd8a*, *T-bet*, *C3*, *il12a*, *il12b* and *cd79a* also showed different expression patterns during seahorses' pregnancy, further indicating the complex and unique immune features of male pregnancy (Supplementary Fig. 29). In addition, an expanded comparative approach revealed that asplenia and male pregnancy characters evolved independently but cooccurred on the same branches (*Hippocampus* and *Syngnathus*) (Supplementary Fig. 30).

## Discussion

Seahorses possess the most advanced form of full internal male pregnancy compared to other syngnathids, and simultaneously, exhibit asplenia. A pregnant male seahorse faces the dilemma to immunologically defend both itself and the embryo against prevailing pathogens, while concurrently the semi-allogenic embryo has to be immunologically tolerated[41]. Pipefishes and seahorses adapted their immune biology encompassing modifications and losses of immune-related genes, e.g. CD8b and MHC II pathway genes, which illustrates the remarkable flexibility of the vertebrate immune system in general[6]. Pregnant males further change the expression of immune genes like *ptgs2*, *pla2g4a*, *chia* and *b2m*[6,42] presumably supporting the immunological tolerance of the embryo. As a vital part of the adaptive immune system, the lack of the MHC class II pathway apparently does not

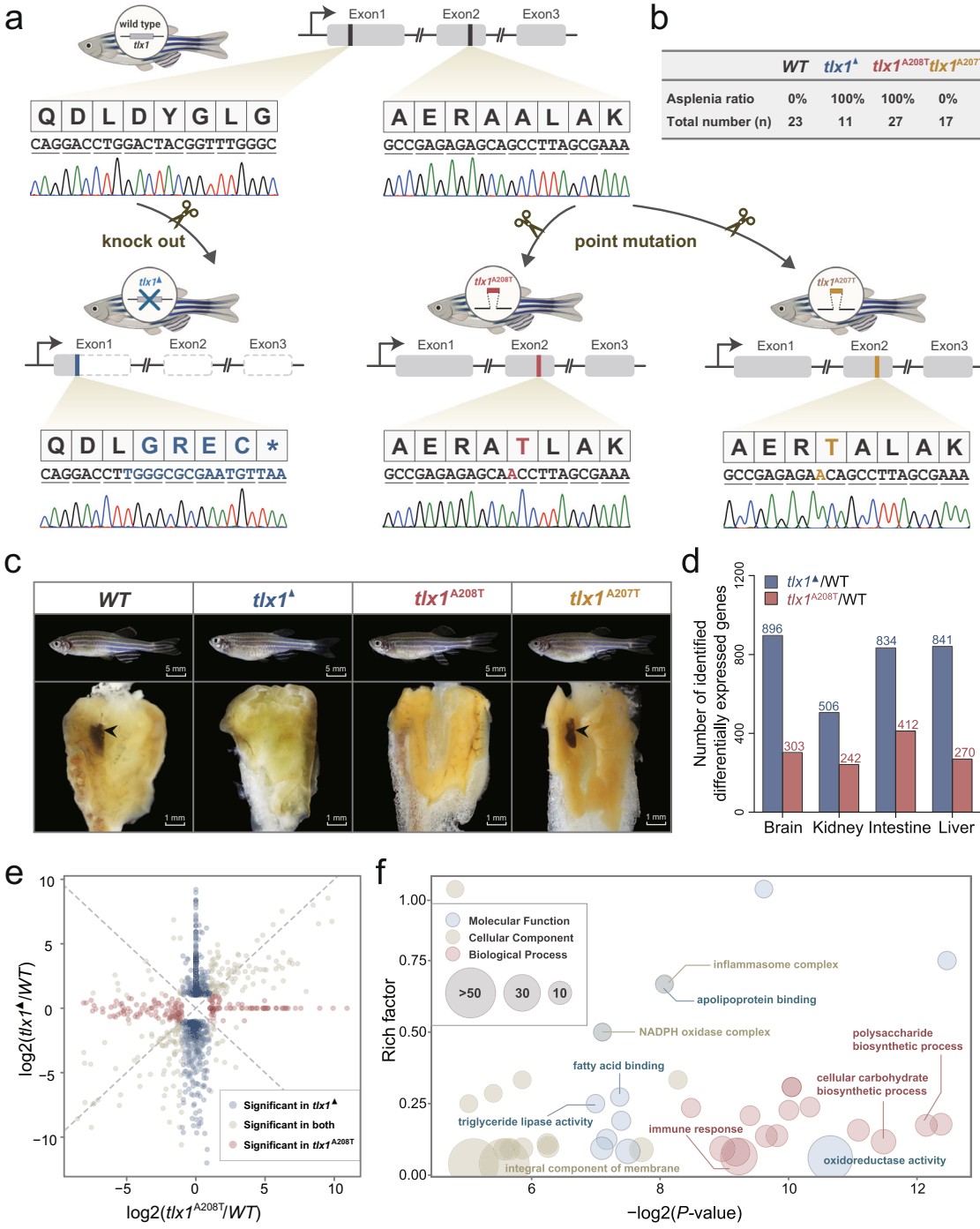

**Fig. 3 | Missense mutation of *tlx1* controls asplenia phenotype in zebrafish.**
**a** Genome editing of three zebrafish lines (*tlx1*▲, *tlx1*A208T and *tlx1*A207T) using CRISPR/Cas9 technology. Procedures are detailed in Supplementary Fig. 15. ▲ represents full knockout of the gene, while A208T and A207T indicate the specific point mutations at sites 622 and 619 of zebrafish *tlx1*, respectively. Blue, *tlx1*▲; Red, *tlx1*A208T; Yellow, *tlx1*A207T. The *tlx1* nucleotide sequence of genomic DNA of three zebrafish lines was validated by Sanger sequencing. The figure was created with BioRender.com.
**b**, **c** *tlx1*▲ and *tlx1*A208T zebrafish exhibited asplenia while *tlx1*A207T zebrafish had an intact spleen; the asplenia ratio of individuals was calculated and listed (*n* = 11–23). The splenic phenotype of other 74 individuals were listed in Supplementary Figs. 16–18. WT wild type. **d** Differentially expressed gene (DEG) number of *tlx1*▲

and *tlx1*A208T compared to that of wild-type individuals via comparisons of the brain-, kidney-, intestine-, and liver-transcriptomes. **e** Volcano map of the shared and group-specific DEGs in the brain of *tlx1*▲ and *tlx1*A208T compared to that in the wild-type group. Blue, red, and gray represent significantly changed DEGs in *tlx1*▲, *tlx1*A208T, and both groups, respectively. **f** GO enrichment of the DEGs between *tlx1*▲ and *tlx1*A208T illustrate the different in gene expression patterns associated with brain development and function. The enrichment was conducted using the GOseq R package, and corrected *P* < 0.05 indicated significant enrichment. The size of the circle represents the number of genes in each category. Source data are provided as a Source Data file.

always lead to asplenia, since spleens are present in other species that lost MHCII, as is known from Gadiformes (e.g., Atlantic cod) and the non-parasitizing anglerfish *Lophius piscatorius*[43–46]. In addition, as an important secondary lymphoid organ that performs mostly

immunological functions, the loss of a spleen - mediated by knockout of *tlx1* - is linked to an immunological deficiency in mice and zebrafish[19,24,47]. Our results strongly suggest that a seahorse-specific mutation in *tlx1* leads to asplenia, as confirmed by CRISPR-Cas9

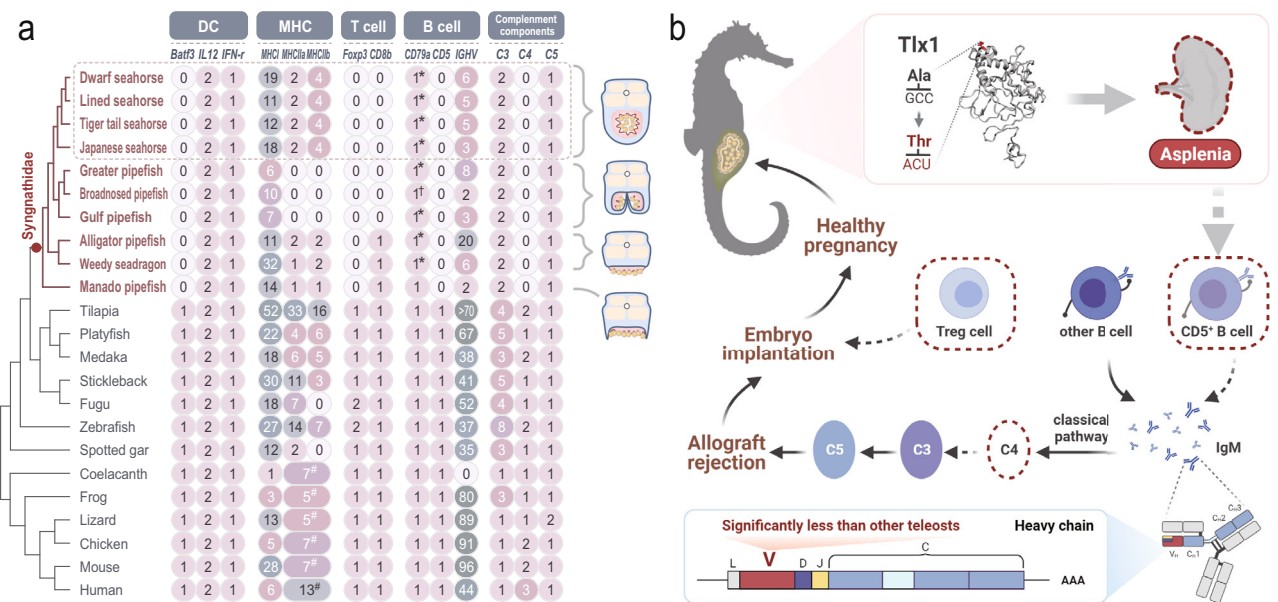

**Fig. 4 | Immunogenomic basis of asplenia and male pregnancy in seahorses.**
**a** Copy numbers of key genes involved in the development and function of DC, MHC, T/B lymphocytes, and complement components 3–4 in seahorses and other vertebrates. DC dendritic cell, MHC major histocompatibility complex. *, variation of ITAM (immunoreceptor tyrosine-based activation motif) region; †, based on partial sequence (low-coverage sequencing and genome assemblies), but featuring the variation of ITAM; #, Total number of MHC II. The brood pouch types are shown on the right. *Hippocampus* and *Syngnathus* species exhibit the closed brood pouches. **b** Schematic map illustrating the hypothesized molecular trade-offs in the male pregnancy of seahorses. Asplenia caused by the *tlx1* mutation could decrease the CD5+ B cell populations. In addition to the corresponding reduced diversity in antibodies, the classical complement pathway triggered by antibody-antigen-interactions might also be weakened, affecting the downstream events usually triggering allograft rejection. Loss of *C4* is expected to be associated with decreased abundance of Treg cells, which orchestrate self-immune tolerance. The shaded icon and dotted line indicate gene loss in seahorse genomes. Treg cell, regulatory T cell. The figures were created with BioRender.com.

experiments. This finding confirms that *tlx1* is part of a non-redundant transcription factor network required for the development of the spleen, and also attests to the evolutionary stability of transcription factor networks in regulating organ development between teleosts and mammals.

Immunological adaptations are key in the evolution of extraordinary reproduction strategies. Female pregnancy evolved convergently over 100 times in vertebrates and coevolved by fine-tuning of the immune system. Similarly, in deep-sea angler fishes, the loss of parts of their immune system crucially permitted the integration of the male body into the female body[48]. Male pregnancy has evolved along a gradient of increasing parental care complexity, ranging from simple egg attachment to the males' ventral side (e.g., *Nerophina*) to the complexity seen in *Syngnathus* and *Hippocampus* that evolved brood pouches with intricate placenta-like structures. The loss of the identified immunogenomic features in syngnathid fishes emerged concomitantly with the evolution of increasing parental care complexity, as was previously shown for the loss of MHC II and *cd8b*[6]. Combined with the here identified loss of *batf3*, this suggests a modification of antigen processing and antigen-specific immunity in seahorses. Alternatively, however, seahorse male pregnancy may also rely on alternative dendritic cell regulation via *batf*, *il12* and *ifng* - genes which are present in all syngnathids, as *cd8a* likely compensates for the loss of its paralog[6]. Coincidentally, a previous study on single-cell transcriptomics of immune cells in *Syngnathus typhle* have identified a number of MHC I-related pathway constituents and cytotoxic-related genes, which supports the presence of the CD8 + T-cell subset in pipefish[49]. The reduced copy number of *C3* and the loss of *C4* genes in seahorses further suggest an impeded activation of the complement system, which can reduce antibody-mediated allograft recognition and inflammation. Along the same lines, our current data add to the hypothesis that syngnathid immune biology might evolve synchronously with their unique male pregnancy[16].

The main responsibility for the transient T cell tolerance specific for paternal antigens during pregnancy was assigned to regulatory T cells (Tregs), as these enhance immune tolerance during pregnancy[39]. Asplenia might also be directly or indirectly linked to the loss of *foxp3*, as splenectomy patients exhibit a decrease of blood FOXP3+ Treg cells[40], implying a potential link between the spleen and the abundance of FOXP3+ Treg cells. In addition, sex-dependent immunological differences affect immune responses to both self and foreign antigens, and Treg numbers are associated not only with sex steroid levels but also with sex[50]. The loss of *foxp3* indicates that seahorses seem to have adopted a unique evolutionary strategy for their unparalleled sex-role reversal brooding and nourishing paternal care. Indeed, as a conserved organ, the loss of the spleen may not only be accompanied by the evolution of male pregnancy but may also bring about changes in some other traits, such as the observed immune response changes (including the lost/contracted genes like *batf3*, *C4*, *C3*, *CD5*, and *foxp3*). Hence, our speculation that the synchronized evolution of asplenia and male pregnancy may not be accidental but instead an evolutionary link exists. In addition, as demonstrated here, this reduction in the immune repertoire is shared across the family *Syngnathidae* and is not unique to seahorses; thus, further analyses will be necessary to evaluate the immune response strategy in species with different levels of pregnancy complexity in the future.

Seahorses represent a single lineage within the Syngnathidae, a family with >350 species of which many show less derived forms of male pregnancy - potentially facing pregnancy-related immune challenges that seahorses seemingly have overcome. Moreover, much of the syngnathids' diversity still remains unexplored to date and the variations in morphological and physiological complexity of immune and brooding organs have only been described superficially[9]. Functional redundancy in the immune system makes it hard to predict the consequences that gene loss might have on an organism. However, it is also this redundancy that provides the evolutionary opportunity to

organisms to tinker with the regulatory pathways orchestrating an effective immune response while accommodating for physiological challenges, such as pregnancy, resulting in an unexpected genomic flexibility of vertebrate immune systems. For now our understanding of the intricate immune challenges syngnathids face at different stages of male pregnancy is still incomplete, our study sheds light on the genetic basis of a major adaptive immunological novelty putatively associated with the evolution of sex-role reversal and male pregnancy in seahorses. Considering the entire syngnathid phylogeny, there is still deep divergence between the *Hippocampus* and *Syngnathus* lineages[23], and the splenic phenotype and *tlx1* gene sequence information of lineages more closely related to *Hippocampus*, which are missing in our study, would improve the resolution of the dataset and thus allow us to formulate better informed conclusions. Therefore, we encourage further studies filling these gaps in genomic knowledge.

## Methods

### Animal use ethics

All animal experiments were conducted per the guidelines and approval of the respective Animal Research and Ethics Committees of the South China Sea Institute of Oceanology, Chinese Academy of Sciences.

### *Hippocampus* spp. specimens and nucleic acid sample preparation

Two individuals, one adult male Japanese seahorse *H. mohnikei* and one adult male dwarf seahorse *H. zosterae*, were used for whole-genome sequencing in this study. *H. mohnikei* was collected from local markets in Rizhao (Shandong, P.R. China) in 2017. *H. zosterae* was donated by aquariums in Guangzhou to the Marine Biodiversity Collections of South China Sea, Chinese Academy of Science in 2015 (Supplementary Table 1). Genomic DNA was extracted from each sample using a standard phenol-chloroform protocol.

### Genome sequencing and Hi-C sequencing

Two paired-end libraries with 350 bp insert size were constructed for seahorses using the Illumina HiSeq 2500 system (San Diego, USA). The quality-filtered reads were used for genome size estimation via the *K-mer* method[51]. The genome size was then estimated as follows: Genome Size = $K_{num}/K_{depth}$. We also calculated and plotted the 19-mer depth distribution. In addition, Nanopore sequencing was performed for *H. mohnikei*. Briefly, libraries were constructed and sequenced on R9.4 FlowCells using the MinION sequencer (ONT, UK). All sequencing was performed by BioMarker Technologies Company (Beijing, China). PacBio sequencing was performed for *H. zosterae*. The genomic DNA of the sample was sheared to an average size of 20 Kb using a g-TUBE device (Covaris, Woburn, MA, USA). The sheared DNA was purified and end-repaired using polishing enzymes, followed by blunt end ligation reaction and exonuclease treatment to create a SMRTbell template according to the PacBio 20-kb template preparation protocol. A BluePippin device (Sage Science, Beverly, USA) was used to size-select the SMRTbell template and enrich large (>10 Kbp) fragments. Single-molecule sequencing was then conducted on a PacBio Sequel platform to generate long-read data.

A blood sample of the male *H. zosterae* was also used for Hi-C sequencing by the Illumina HiSeq 2500 platform, using paired-end of 150-bp reads. DNA was isolated from the sample and the fixed chromatin was digested with the restriction enzyme DpnII overnight. Subsequently, the DNA was sheared by sonication to the mean size of 350 bp. Hi-C libraries were generated using NEBNext Ultra enzymes and Illumina-compatible adaptors. Biotin-containing fragments were isolated using streptavidin beads. All libraries were quantified by Qubit2.0, and the insert size was checked using an Agilent 2100. The libraries were then quantified by qPCR. The Hi-C data were mapped to PacBio-based contigs using BWA (version 0.7.10-r789; mapping

method: aln). Uniquely mapped data were used for chromosome-level scaffolding. HiC-Pro (version 2.8.1)[52] was used for duplicate removal and quality controls, and the remaining reads were valid interaction pairs for further assembly.

### Genome assembly

PacBio subreads were corrected and trimmed using Canu (version 1.5, available at https://github.com/marbl/canu)[53]. wtdbg generated a draft assembly with the command 'wtdbg -i pbreads.fasta -t 40 -H -k 21 -S 1.02 -e 3 -o wtdbg' using error-corrected reads from Canu. A consensus assembly was obtained with the command 'wtdbg-cns -t 40 -i wtdbg.ctg.lay -o wtdbg.ctg.lay.fa -k 15'. Next, Illumina paired-end reads were also aligned for consensus assembly using BWA (version 0.7.10-r789; mapping method: MEM) and the polishing step was performed with the Pilon (version 1.22) software, with the following parameters: --mindepth 10 --changes --threads 4 --fix bases. The polishing step was iterated two times. The assembly statistics are shown in Supplementary Table 5. Benchmarking Universal Single-Copy Orthologs (BUSCOs) assessment showed that our assembly captured 93.46% and 92.95% of complete BUSCOs of *H. mohnikei* and *H. zosterae*, respectively.

Using contigs assembled from the PacBio data, Hi-C data were used to correct misjoins in contigs, order, and orient contigs. Pre-assembly was performed for contig correction by splitting the contigs into segments with an average length of 300 kb. The segments were then preassembled with Hi-C data. Misassembled points were defined and broken when split segments could not be placed to the original position. Next, the corrected contigs were assembled using LACHESIS with parameters CLUSTER_MIN_RE_SITES = 225, CLUSTER_MAX_LINK_DENSITY = 2, ORDER_MIN_N_RES_IN_TRUN = 105, and ORDER_MIN_N_RES_IN_SHREDS = 105 with Hi-C valid pairs. Gaps between ordered contigs were filled with 100 'N's. The sequence interaction matrices are shown in Supplementary Fig. 2, and the statistical analysis results of chromosome assemblies are summarized in Supplementary Table 6.

### Genome annotation

De novo identification of repeats and transposable elements were performed using the PILER-DF[54] and the RepeatScout[55] under default parameters. RepeatMasker and RepeatProteinMask (version 3.3.0) were employed to identify transposable elements (TEs) based on homology searches against the Repbase library (release 16.03) using the parameters "-nolow -no_is -norna -parallel 1" and "-noLowSimple –pvalue 1e-4". Ab initio gene prediction was performed using three programs, namely Augustus[56], GlimmerHMM[57], and SNAP[58]. The GeMoMa[59] program was run for homology-based prediction by aligning the assembled genome against *O. latipes*, *S. salar*, *H. comes*, and *H. erectus*. Next, the transcriptome data of *H. erectus* from our previous work[16] were downloaded and mapped onto the genome, and gene prediction was performed by TransDecoder (http://transdecoder.github.io) and GeneMarkS-T. PASA[60] was used to predict the Unigene sequences without reference assembly based on transcriptome data. We then combined the results from these three methods with EvidenceModeler[61]. Gene functions were further annotated by searching publicly available databases, including the NR, KOG, KEGG, GO, and TrEMBL databases.

### Phylogenetic analysis

Protein datasets were obtained from Ensembl-FTP release-96 [*T. rubripes*, *G. aculeatus*, *O. latipes*, *O. niloticus*, *P. magnuspinnatus*, *D. rerio*, *X. maculatus*, and *L. oculatus*] or other sources [*G. morhua*[62], *H. comes*[16], *H erectus*[17], *H. zosterae*, and *H. mohnikei* (present study)]. One-to-one orthologues were identified using OrthoFinder version 2.2.7 at default settings from the 13 ray-finned fish species. Protein sequences for these one-to-one orthologues were then extracted into their

respective orthogroups using an in-house Perl script. Multiple alignments were generated for each of the orthogroups using MAFFT (version 7.475). The individual protein alignments were concatenated together using an in-house Perl script. A trimmed concatenated alignment was generated using Gblocks 0.91b with the 'allowed gap positions' set to "With Half". ModelFinder[63] was used to deduce the best-suited substitution model for the trimmed alignment (JTT + F + I + G4 model). For the maximum likelihood analysis, we employed the best fit substitution model as deduced by ModelFinder and 1000 replicates using the ultrafast bootstrap approximation approach[64] as implemented in IQ-TREE version 1.6.10.

## Expansion and contraction of gene families

Expansion and contraction in gene families were calculated by the CAFÉ program (v 3.1) based on the birth-and-death model[65]. The parameters "-p 0.01, -r 10000, -s" were set to search the birth and death parameter (λ) of genes based on a Monte Carlo resampling procedure, and birth and death parameters in gene families with a $P$-value ≤ 0.01 have been reported. The gene families without homology in the SWISS-PROT database were filtered out to reduce the potential false positive expansions or contractions caused by gene prediction. Go and KEGG terms of all proteins used in the comparative analysis were annotated with eggnog 5.0[66]. The gene family with more than 90% of its members sharing the same annotations was considered as the single functional family and its weighting was set to 1. For gene families containing sequences having multiple functional annotations, different weighting values were assigned to each functional annotations according to the ratio of the annotated times of each term to the total annotated times of all members. The total weighting value were 1 for each family.

## Gene loss

When a gene has no homologs within the seahorse clade, but the homologs of that gene are present in the closest sister lineage of the seahorse clade, we consider that the gene was lost in seahorses[67]. One-to-one orthologous genes were extracted from each species, and multiple sequence alignments were generated accordingly. The gene loss analysis was performed using an in-house script in $R$.

## Positive selection of genes (PSGs)

We tested for PSGs in the seahorse lineages compared to all other background species. Orthologous genes were extracted from the same species selected for the gene family expansion/contraction analysis. Multiple sequence alignments were generated using the MUSCLE software (v3.8.31). Positive selection analyses were conducted with the branch-site model using PAML[68]. We compared model A (allows sites to be under positive selection; fix_omega = 0) with the null model A1 (sites may evolve neutrally or under purifying selection; fix_omega = 1 and omega = 1) via likelihood ratio test using the Codeml program in PAML. The significance of the compared likelihood ratios was evaluated by χ2 tests from PAML. Then the p.adjust function embedded in the R language was used to adjust the $P$ value using all the original $p$-values. Only the genes with an FDR-corrected $P$ value smaller than 0.05 were considered positively selected.

## Rapidly evolving genes (REGs)

The same orthologous genes and tree topology as those for PSGs were used in the analysis. The branch model in PAML was used, with the free-ratio model (model = 1) allowing the value to vary on each branch. The dN values of all nodes and tips on each tree were tested using the t.test function in R programming and then the $p$-values were adusted using FDR. Genes with an FDR-corrected $P$ value smaller than 0.05 and the dN of the seahorse lineage higher than that of the sister lineage of seahorse were considered as rapidly evolved on seahorse lineage. GO and KEGG enrichment analyses of REGs were then Performed.

## Lineage-specific mutated genes (LSGs)

To identify LSGs in the seahorses, the same orthologous genes and tree topology produced from the PSGs analyses were used. Only the ancestral state of the seahorse species was different from that of all other background species genes were recognized as the true LSGs. To ensure result reliability, only ancestral amino acids with posterior probabilities ≥ 0.95 were used to determine lineage-specific mutation. Moreover, we discarded lineage-specific mutated sites within poorly aligned fragments: we calculated the pairwise sequence similarities of 10 amino acids centering each lineage-specific mutated site. If the mean similarity was lower than 0.7 or the lowest was lower than 0.35, this fragment was defined as a poorly aligned fragment[69]. GO and KEGG enrichment analyses of LSGs were then performed.

## Local gene synteny and phylogenetic analysis of the *tlx* gene families of the lined seahorse

Syntenic genes in other species (fruit fly, amphioxus, human, and zebrafish) were acquired using the Ensembl genome browser from Genomicus (https://www.genomicus.biologie. ens.fr/genomicus-100.01). The output was sorted for genomic contig/scaffold/linkage and then for start position within each of these genes. Next, phylogenetic analysis based on Tlx1, Tlx2, and Tlx3 amino acids sequences was performed in vertebrates including lined seahorse. A total of 41 species of cyclostomes, fishes (26 species from 16 orders), amphibians, reptiles and mammals were included. Protein sequences were aligned using MAFFT (version 7.475) with minor manual adjustments. A maximum-likelihood tree was generated with IQtree v.2. Boot-strap support was established through 1000 iterations for Ultra-Fast Bootstrapping and SH-like approximate likelihood ratio test. The consensus tree was visualized by the FigTree (version 1.4.4) software.

## Alignment of Tlx1 amino acids in different lineages

Next, Tlx1 amino acid sequences from representative vertebrates were compared using Clustal W 2.1. The GenBank accession numbers of these genes are listed in Supplementary Table 16. The results showed that the amino acid sequences of the homeobox were completely identical among all vertebrates, except for the seahorse-specific mutation in which the hydrophilic threonine (T) replaced the hydrophobic alanine (A) exclusively in the seahorse lineage (Supplementary Fig. 10a). Next, we manually Extended Data the scope of comparison to a broader taxa selection, including mammals, birds, reptiles, amphibians, and fishes. A total of 38 vertebrates containing 23 fish species from 15 orders were included, and we confirmed that among these species, only seahorses harbored the lineage-specific mutation (Supplementary Fig. 11a).

## Validation of Tlx1 amino acids in 18 seahorse species and other Syngnathidae fishes

Further PCR analysis was conducted using a total of 18 seahorse species to validate this lineage-specific mutation. Genomic DNA was extracted from 18 seahorse species, namely *H. abdominalis, H. algiricus, H. barbouri, H. camelopardalis, H. capensis, H. comes, H. erectus, H. fuscus, H. hippocampus, H. histrix, H. ingens, H. jayakari, H. kelloggi, H. kuda, H. mohnikei, H. reidi, H. spinosissimus,* and *H. zosterae*. The primers (F: 5′-TCTCACCGCTCACTGTAAC-3′; R: 5′-GGTCATTT TGAGGGCTTT-3′) were designed to amplify the second exon of *tlx1* in these seahorses. The PCR procedure was conducted using standard PCR conditions. The PCR products were loaded on a 1.5% agarose gel and electrophoresed at 130 V for 25 min. The results were photographed under a UV light. We also investigated the mutation locus by comparison with the genomes of all other available members of the Syngnathinae subfamily, including *S. acus, S. typhle, S. scovelli, S. biaculeatus, P. taeniolatus, O. manadensis* and *Nerophis ophidion*.

## The splenic phenotype of the Syngnathidae

To clarify the splenic phenotype of the Syngnathidaes, we detected the most available samples (including *H. abdominalis*, *H. erectus*, *S. typhle*, *S. biaculeatus* and *Nerophis ophidion*). Briefly, after euthanasia with 300 ng/ml MS-222 (Sigma-Aldrich, USA), the individuals were dissected and imaged by a MVX10 camera (Olympus, Japan) or Olympus SZX2 Stereo Microscopes (Olympus, Japan). In addition, the Micro CT scan of the *S. biaculeatus* was conducted on the Nemo Micro-CT (NMC-200) of PINGSENG Healthcare Inc, a high-resolution imaging technology based on cone beam the CT principle. Firstly, samples were soaked in the contrast agent for 12 days and then were placed into the sample chamber vertically, whose scanning tube voltage and current were set to 90 kV and 90uA, respectively. During the scanning process, the detector and bulb rotated 360° around the central axis of the sample chamber and performed 10,000 projections in the scanning area, which lasted 1000 s. After capturing images by the detector, it was reversely reconstructed using the FDK method on Avatar software (version 1.7.2, PINGSENG Healthcare Inc.), with a pixel size of 10 um × 10 um × 16 um. Histological analyses of the *S. biaculeatus* spleens were conducted, then the transcriptomic profiles of the *S. biaculeatus* spleens and the *H. erectus* small white organ were also sampled and sequenced (Supplementary Data 9).

## Generation of *tlx1*▲ zebrafish lines

Strain *AB zebrafish (*D. rerio*) were maintained under a 14-h light (8:00-22:00)/10-h dark (22:00-8:00) cycle at 26–28 °C to induce spawning. CRISPR/Cas9-mediated mutagenesis was used for generating indels in zebrafish *tlx1* (ENSDARG00000003965; http://ensembl.org) using sites identified by ZiFiT Targeter (http://zifit.partners.org/ZiFiT/). Mutagenesis targeting two regions in *tlx1* exon 1, GGACCTGGAC-TACGGTTT (site 1) and GGTCCTACAACATGAACTT (site 2), was induced. Purified gRNAs (~80 pg) were co-injected with Cas9 mRNA (~400 pg) into zebrafish embryos (F0 fish) at the one-cell stage. Two days after injection, 8–10 embryos were collected for genomic DNA extraction to check whether the targeted genomic fragment was mutated. The target genomic regions were amplified by PCR and subcloned into the pTZ57R/T vector. The adult founders were outcrossed with wild-type fish to obtain F1 fish, which were subsequently genotyped and outcrossed with wild-type fish to yield F2 fish. Next, heterozygous F2 individuals were intercrossed to produce homozygous F3 fish. Finally, we generated two *tlx1* knockout zebrafish lines. Two *tlx1* nonsense alleles with 122-bp deletion *tlx1*▲ (−122) and 5-bp deletion *tlx1*▲ (−5) in the first exon were generated (the red arrowheads and the yellow-backed sequence), which caused frame-shift mutations at the position 43 and 47 AA, respectively (the red-backed sequence), as well as premature transcription termination event at the position 47 and 58 AA (the green-backed sequence) (Supplementary Data 10).

## Generation of *tlx1*^A208T and *tlx1*^A207T zebrafish lines

CRISPR/Cas9-mediated homologous recombination (HR) was used for generating point mutations in zebrafish *tlx1*. We also used sites identified by ZiFiT Targeter (http://zifit.partners.org/ZiFiT/) to design CRISPR/Cas9 targets near the point mutation sites. The designed sites targeted two regions in *tlx1* exon 2, GGCGCGTGAACGACGTCCG (site 3) and GGAGGTGTTCGGTTCTGGTA (site 4). HR donor plasmids were constructed using the Hieff Clone Plus Multi One Step Cloning Kit (YEASEN, China). The G622A HR donor was constructed by ligating four fragments (a left arm, a middle arm, a selective marker, and a right arm) with the pHRHG vector (XinJia, China). The fragments of the left arm (1081 bps), middle arm (923 bp), and right arm (950 bp) were amplified from the genomic DNA of AB/WT zebrafish by using the PrimeSTAR HS DNA polymerase (Takara, Japan). The point mutation (G622A) in the middle arm was introduced using the Fast Mutagenesis System (Transgen, China). To prevent the HR donor from being excised by the CRISPR/Cas9 system, the silent mutations of the two

CRISPR/Cas9 target sites in the middle arm was also introduced using the Fast Mutagenesis System (Transgen). The selective marker was amplified from a heart-specific transgene plasmid including *myl7* promoter, DsRed and SV40 polyA signal. Furthermore, *frt* sites were added to both sides of the selective marker, which could be removed through excision by Flp recombinase. The G619A HR donor was constructed similarly.

The G622A donor plasmid was purified before microinjection by the Gel Extraction Kit (Qiagen). The zCas9 protein, sgRNAs, and donor plasmid (15 pg) were co-injected into zebrafish embryos at the one-cell stage. Adult F0 zebrafish were outcrossed with wild-type fish to screen the founders with selective marker expression and correct genotyping. The founders were then outcrossed with wild-type fish to yield F2 fish, in which the selective marker was excised through injection of *flp* mRNA (50 pg) at the one-cell stage. Subsequently, heterozygous F2 individuals without selective marker were intercrossed to produce homozygous F3 fish. And the G619A line was also obtained similar to the G622A line. The primers were listed in Supplementary Data 10.

## Genotyping and splenic phenotype detection

Zebrafish were euthanized with 200 ng/ml MS-222 (Sigma-Aldrich, USA) and genotyped via tail fin clipping. Genomic DNA was extracted from the fin tissues. Genotypes of the three mutation lines (*tlx1*▲, *tlx1*^A208T and *tlx1*^A207T) were determined using standard PCR conditions and then sequenced by Sanger sequencing (Sangon Biotech Co., Ltd, China). The primers were as follows: *tlx1*▲, F: 5′-ATCGTCTGTAGT TCCGTCTTTC-3′, R: 5′-ACCGCTTTAACCGCTGAGA-3′; *tlx1*^A208T and *tlx1*^A207T, F: 5′-ATTTGCCTTCCACTGCTTGG-3′, R: 5′-CCAGCTGACCT-CACGGTTTAT-3′.

Whole-mounts of abdominal organs were carefully removed in accordance with a previous study by Xie et al.[24]. Briefly, after euthanasia with 200 ng/ml MS-222 (Sigma-Aldrich, USA), the whole-mounts of abdominal organs were carefully removed from wild-type (23 fish) zebrafish as well as *tlx1*▲ (11 fish), *tlx1*^A208T (27 fish), and *tlx1*^A207T (17 fish) mutants (Fig. 3). The organs were then fixed in 4% PFA for 5 min. All zebrafish checked were sampled randomly and done blind with respect to mutant status. Representative images were obtained using a MVX10 camera (Olympus, Japan) (Fig. 3 and Supplementary Figs. 16–18).

## Whole mount in situ hybridization

Lined seahorse embryos at different developmental stages (early stage (S1), ~6 days post fertilization, dpf; mid stage (S2), ~12 dpf; late stage (S3), ~18 dpf) were sampled and fixed in 4% paraformaldehyde (PFA)/ phosphate-buffered saline (PBS) overnight at 4 °C, in accordance with the methods of Leerberg et al.[70]. Primers were designed for hybridization of the *tlx1* gene in lined seahorse (F: 5′-GATCACATGGGAC TAGCGGCAC −3′; R: 5′- GGCGTAATACGACTCACTATAGGGGTGTTACA GTGAGCGGTGAGAGAG−3′). The RNA antisense probes were then yielded through in vitro transcription using the DIG RNA Labeling Kit (SP6/T7) (Roche, Mannheim, Germany). During hybridization, the seahorse embryos were digested with 10 µg/ml proteinase K solution for an appropriate time and subsequently fixed again in 4% PFA for 20 min. Next, the embryos were preincubated in antibody blocking solution for 1 h at 20 °C with PBSTw and then incubated for 2 h at 20 °C in 1:3000 dilution of anti-DIG-AP Fab. The samples were washed twice for 5 min each in PBSTw and developed in coloration solution (30 µl NBT/BCIP stock in 10 ml Coloration buffer) at 4 °C. The reaction was terminated by washing with 50% and 100% methanol for 5 min each. The embryos were then fixed with 4% PFA overnight in darkness, mounted in 100% glycerol, and imaged under an Olympus SZX2 Stereo Microscopes (Olympus, Japan).

## Transcriptome analyses

Transcriptome analysis of the brain and three immune-related organs (the liver, kidney, and intestine) of zebrafish. RNA sequencing libraries

of these tissues (each tissue including four biological replicates) of the wild-type, *tlx1*▲, and *tlx1*[A208T] zebrafishes were constructed, and then sequenced on a flow cell using an Illumina HiSeq™ 2500 platform (Supplementary Table 17). Raw data (raw reads) of FASTQ format of these four tissues were first processed using in-house Perl scripts. Clean reads were mapped to the zebrafish genome assembly (GRCz11) using the TopHat2 program. Gene expression levels were estimated with fragments per kilo-base of exon per million fragments (FPKM values) using the Cufflinks program. Differential expression genes (DEGs) were checked using DEGSeq2 and identified based on corrected *p*-values (*Q*-value) and false discovery rates. Pair-wise comparisons across 12 combinations (brain: *tlx1*▲ vs. WT, *tlx1*[A208T] vs. WT, *tlx1*▲ vs. *tlx1*[A208T]; kidney: *tlx1*▲ vs. WT, *tlx1*[A208T] vs. WT, *tlx1*▲ vs. *tlx1*[A208T]; liver: *tlx1*▲ vs. WT, *tlx1*[A208T] vs. WT, *tlx1*▲ vs. *tlx1*[A208T]; intestine: *tlx1*▲ vs. WT, *tlx1*[A208T] vs. WT, *tlx1*▲ vs. *tlx1*[A208T]) were conducted. Only genes with an absolute value of log2 (fold change) ≥1 and false discovery rate significance score <0.05 were used for subsequent analysis. GO enrichment in four tissue combinations of *tlx1*▲ vs. *tlx1*[A208T] were conducted using the GOseq R package, and corrected *P* < 0.05 indicated significant enrichment.

### Comparative whole genome search for seahorse immune/pregnancy genes

We scanned the genomes for a subset of immune-related genes that might be involved in pregnancy in four seahorse species, as well as six other Syngnathidae species, including greater pipefish, broadnosed pipefish, gulf pipefish, alligator pipefish, weedy seadragon, and Manado pipefish. The query protein sequences of these genes were obtained from Ensembl Genome Browser 104 or NCBI with representatives from *Homo sapiens*, *T. rubripes*, *O. latipes*, and *G. aculeatus*. Firstly, the genomes of the 10 Syngnathidae species were compared with those of other vertebrates using the best-hit search in TBLASTN v2.9.0 to find matching reads in our genomic datasets. BLAST parameters were set to an expectation cutoff of 1E − 5, allowing a maximum number of 1,000 returned sequences. In addition, all above immune-related genes were predicted using the homology-based gene prediction tool GeMoMa v1.7.1[59,71], with four species as reference organisms. The selected species were zebrafish, medaka, fugu, and stickleback. Introns from the mapped RNA-seq reads were extracted and filtered by the GeMoMa modules ERE and DenoiseIntrons. Next, we independently ran the module GeMoMa pipeline for each reference species using MMseqs2[72] as alignment tools, on the mapped RNA-seq data. Finally, the eleven individual annotation results were combined into a final annotation by using the GeMoMa modules GAF and AnnotationFinalizer[73]. Moreover, for immunoglobulin heavy chain genes (*ighvs*), the genome of the tiger tail seahorse has been fully predicted in Ensembl. Taking the sequence of the tiger tail seahorse as a reference, we used Exonerate (version 2.2.0) to conduct homology-based gene prediction for other genomic datasets with the default parameters. Phylogenetic analyses of genes identified in seahorses and other representative vertebrates were conducted using the amino acid sequences. The GenBank accession numbers of these gene families used for phylogenetic analysis are listed in Supplementary Data 13.

### Reconstruction of character state evolution

Seven non-Syngnathiformes *L. oculatus*, *D. rerio*, *T. rubripes*, *G. aculeatus*, *O. niloticus*, *O. latipes*, *X. maculatus*, and thirteen Syngnathiformes including *H. comes*, *H. erectus*, *H. zosterae*, *H. mohnikei*, *S. acus*, *S. typhle*, *S. scovelli*, *P. taeniolatus*, *S. biaculeatus*, *O. manadensis*, *N. ophidion*, *F. commersonii* and *A. strigatus* were used to conduct the characters' reconstruction analysis based on previous study[7]. Briefly, the phylogenetic reconstruction of these species was conducted with the methods in the section phylogenetic analysis with minor modifications. Available empirical data on four characters including amino acid replacement in *tlx1*, brood pouch development, presence of spleen and immune genes simplification for each species included in the phylogenetic analyses were mapped onto our molecular phylogeny in Mesquite Ver.3.70 (http://www.mesquiteproject.org/).

### Real-time quantitative PCR of genes involved in pregnancy

Adult male non-pregnant (*n* = 4) and pregnant (*n* = 4) lined seahorses collected from a fish farm (Zhangzhou, Fujian, China) were anesthetized with MS222 before brood pouches sampling. After the separation of embryos, the brood pouch was washed with PBS to remove embryonic contamination and total RNA was extracted using TRIzol reagent according to the manufacturer's instructions. 1 μg total RNA from the brood pouch sample was used to synthesize first-strand cDNA using the ReverAce qPCR RT Master Mix with gDNA Remover (Toyobo, Osaka, Japan). 'No RT' reactions were used as negative controls. The mRNA levels of genes involved in pregnancy, including *C3.2*, *cd8a*, *cd79a*, *gata3*, *il12a*, *il12b*, *T-bet*, and *tgfb1*, were determined by qRT-PCR (Supplementary Fig. 27). The primers used in this study are listed in Supplementary Table 18. qRT-PCR was performed on a Roche Light-Cycler 480 real time PCR system (Roche, Mannheim, Germany), using the SYBR Green I kit (Toyobo, Japan) according to the manufacturer's instructions. *β-actin* was used as an internal control.

### Statistical analysis

Statistical analysis was performed using GraphPad Prism 7.0 (Graph-Pad Software, San Diego, CA, USA). All data are presented as the mean ± standard error of the mean (SEM). Statistical differences were estimated via unpaired Student's *t*-test and the significance level was set at 0.05.

### Reporting summary

Further information on research design is available in the Nature Portfolio Reporting Summary linked to this article.

## Data availability

The whole-genome raw reads and assemblies of *H. zosterae* and *H. mohnikei* have been deposited in the NCBI database under accession code PRJNA797939. The raw reads of the RNA-seq (including *Danio rerio*, *S. biaculeatus* and *H. erectus*) have been deposited in the NCBI database under accession code PRJNA799842. Accessions for previously published genomes used in this study were given in Supplementary Data 15. In addition, the data of the spleens phenotype and histology are available at Figshare (https://figshare.com/projects/Immunogenetic_losses_co-occurred_with_seahorse_male_pregnancy_and_mutation_in_tlx1_accompanied_functional_asplenia/153495). Source data are provided with this paper.

## Code availability

Custom scripts employed for the analysis of the sequencing data are available at Figshare (https://figshare.com/projects/Immunogenetic_losses_co-occurred_with_seahorse_male_pregnancy_and_mutation_in_tlx1_accompanied_functional_asplenia/153495).

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

## Acknowledgements

The authors acknowledge the Marine Biodiversity Collections of South China Sea, CAS for providing samples; the Laboratory Animal Center of Huazhong Agricultural University for providing experimental facilities and the Agency for Science, S. Fan and X. Lu for insightful comments and suggestions that improved the manuscript. This research was supported by the National Natural Science Foundation of China (41890853 to S.Z., 41825013 to Q.L., 42006108 to M.Q.), the Key Research Program of Frontier Sciences of CAS (ZDBS-LY-DQC004 to Q.L.), Guangdong Basic and Applied Basic Research Foundation (2021A1515011380 to Y.L.), the European Research Council (ERC) under the European Union's Horizon research and innovation program (EC/H2020/755659 to O.R.), the Strategic Priority Research Program of CAS (XDB42030204 to Q.L.), and the Key Deployment Project of COMS of CAS (COMS2020Q14 to Y.Z.).

## Author contributions

Q.L., A.M., O.R. and B.V. conceived the project. Q.L. and A.M. supervised the study. X.W. and G.Q. collected the samples. M.Q., R.S., H.Y., X.W., Y.Z., H.Z., Z.Z. and J.Y. performed genome analyses. Y.L., H.Y. and G.Q. performed transcriptomic analysis. Y.L., W.L., B.Z. and H.J. performed CRISPR/Cas9 genomic editing analyses. Y.W. and H.J. performed the Quantitative Real-time PCR. H.J. and Yingyi Z. preformed in situ hybridization. S.Z. performed the Micro-CT analysis. Y.L., M.Q., R.S. and Q.L. wrote the manuscript with input from all other authors. All authors reviewed and contributed to the final manuscript.

## Competing interests

The authors declare no competing interests.
