## [Peer Review File · Nature Communications]

Immunogenetic losses co-occurred with seahorse male pregnancy and mutation in *tlx1* accompanied functional aspleniaReviewers' Comments:

Reviewer #1:

Remarks to the Author:

In this study Liu et al. have compared two high quality genomes of seahorses (generated in this study) with other teleost genomes (previously published) focusing on immune-related genes. The seahorses and the related pipefishes are peculiar because they have evolved male-pregnancy and have undergone severe changes in their adaptive immune system. Furthermore, seahorses and pipefishes within the genus *Syngnathus* have developed closed (male) brooding organs and they also lost their entire spleen (asplenia). Liu and colleagues have found that a single amino acid substitution (Thr – Ala) in the transcription factor *tlx1* is associated with asplenia in seahorses. They further showed by generating CRISPR-Cas9 knockout and point mutation lines in zebrafish lead to loss of spleen in this teleost model system.

This is a well conducted and well-written study providing important comparative immunogenomic results that have been tested experimentally in zebrafish. The results contribute to our understanding of the peculiar phenomenon of male pregnancy, identification of a mutation that leads to loss of spleen as well as immunogenomic modifications in the *Syngnathidae*. I have a few questions for clarifications and some minor comments.

Questions for clarifications:

In the introduction I think it is appropriate to mention that there are two versions of adaptive immunity (MHC and VLR) here or specify that you are referring to the MHC/BCR/TCR system of adaptive immunity (line 45).

Lines 62-63: Does *syngnathids* refer to both pipefishes and seahorses, or just pipefishes? Pipefishes have not been mentioned so far in the text - just seahorses. Maybe just use seahorses here to avoid confusion? – or explain

It is stated in lines 219 – 220 that seahorses and pipefishes have a functional loss of the MHC II pathway. However, in Fig 4 MHC II genes are shown to be present in seahorses (absent in pipefishes). It is referred to Roth et al. 2020 (PNAS). This is still a bit confusing, particularly for a non-pipefish/seahorse scientist – and should be better explained.

In other “systems” where MHC II is lost such as in *Lophiformes* (e.g. monkfish) and *gadiformes* (e.g. Atlantic cod) both these species have spleen (and codfishes in general have spleen, but MHC II is lost from the entire lineage). I think that this issue should be touched upon in the Discussion because there is obviously not a clear connection between loss of spleen and loss of MHC II. Furthermore, according to this ms. it is not clear if the *Syngnathides* with an alanine in *TLX1* actually possess a spleen or not. This issue may also be discussed.

Extended Fig 3b: the Figure shows that the T - A amino acid (point) mutation is not present in all *Syngnathidae*. Indeed, the figure shows *Oostethus manadensis* that display an L in the T/A site as well as T in the adjacent site. As far as I could see this is not commented on, and I am wondering what this could imply.

Minor comments:

Abstract (lines 35-36) it is possible to read this as if the CRISPR-Cas9 experiments were conducted on seahorses and not zebrafish

Line 131: alanine

Line 162: add a short sentence here about what the expression profile is like in wildtype vs asplenia

zebrafish? As of now, the sentence does not provide much insight.

The interesting study by Parker et al. (2022) on single cell transcriptomics of immune cells in pipefish is not mentioned in the Results/Discussion. The results here are relevant to be discussed.

Figure 3: I understand that the knock out/point mutation lines were sequenced by the old Sanger method (perfectly suited for this!) – but it is to this reviewer a bit confusing since sequencing otherwise is done by high throughput methods. Just explain that this is Sanger sequencing in the Fig legend and in Methods.

Reviewer #2:

Remarks to the Author:

Summary:

In this manuscript Liu et al. report on two overarching observations: 1. Loss of suites of genes related to immune system development and function in the seahorse - and with some generality the syngnathid - lineage, and 2. A derived (possibly synapomorphic with respect to a clade that includes seahorses) amino acid substitution in the transcription factor *tlx1*, which is known to influence development of the spleen from previous gene editing studies in mouse and zebrafish. The impetus is that these two patterns may be related (have “coevolved” in the authors’ words), in the context of male pregnancy, although this is the type of cause-and-effect connection that is difficult to establish from observed states in extant organisms, especially with just a single lineage (and evolutionary event), as considered in this study.

To confirm the loss of immune system genes, which some of the authors have reported previously in other papers, the researchers sequenced two additional seahorse genomes and conducted comparative gene family evolution analyses using seahorse genomic resources, several additional syngnathid genomes, and a sample of more distantly related teleost (plus gar) genomes. These genomic resources were also used to identify species with a derived *tlx1* substitution, which include seahorses and members of the *Syngnathus* pipefish genus. The authors then used CRISPR-Cas9 genome editing to knockout *tlx1* in zebrafish, and they also introduced the *Hippocampus/Syngnathus* substitution as a point mutation in a different zebrafish line. According to the authors, both mutations resulted in anatomical asplenia with full penetrance, supporting a role for the gene, and more precisely the specific point mutation, in zebrafish asplenia. They also compare gene expression profiles for various tissues between the two mutants and conclude that the knockout mutant exhibits more transcriptional disruption (more DE genes relative to WT) than the point mutant. The authors report that asplenia is observed in seahorses and *Syngnathus* pipefishes, which suggests the point mutation in zebrafish induces phenocopy, but there are no methodological details, histology, labeling assays, or imaging to describe the syngnathid asplenia phenotype, and the validity of the single supporting reference is questionable (see below).

In general, I view the zebrafish gene editing as main strength of the paper, with the potential to inform on more specific mechanisms by which *tlx1* interacts with other genes to affect spleen development in vertebrates. The primary shortcoming (see below for details) is the lack of convincing support for asplenia in syngnathids, and for direct, systematic relationships between the reported phenotype, immune system gene repertoire evolution, and male pregnancy. For these reasons I think the paper could be a very valuable contribution to the *tlx1* and spleen development literature, published in the appropriate developmental journal, but I cannot recommend publication in Nature Communications.

Primary comments:

1. An especially significant concern is the lack of validation of asplenia in *Hippocampus*, *Syngnathus*,

and other syngnathids. Asplenia is purported by the authors, based on a single reference (Matsunaga and Rahman 1998), which was also the source for the other, secondary citation (Roth et al. 2020). Unfortunately, Matsunaga and Rahman report only that “seahorses might have lost GALT,” based on their inability to identify a spleen from the dissected GI tract of a single, 10 cm *Hippocampus kuda* specimen. No experimental validation is reported in the current manuscript. Because this phenotype (anatomical asplenia) is so essential to the authors’ thesis, a much more rigorous approach, replicated in multiple individuals from multiple syngnathid species, is crucial for establishment as fact. Expectations for this kind of stated finding should probably include more than one assay: options are histology, in situ hybridization with a spleen-diagnostic probe, tomographic imaging, etc.

Particularly important is confirmation of asplenia (or lack thereof) also in the other (non-*Hippocampus/Syngnathus*) syngnathid species for which *tlx1* was examined. For example, if asplenia is confirmed in all syngnathids, then the position 208 A-T substitution can’t fully explain asplenia, because *Syngnathoides*, *Phyllopteryx*, and *Oostethus* all have the ancestral or different derived (*Oostethus*) states. The authors need to convincingly establish asplenia status for all or most of the above lineages to support their conclusions. Because this author group appears to have published on these taxa before, they should have access to the required samples.

2. While the premise of concerted evolution of sweeping loss of immune system genes, loss of the spleen, and complex brooding tissues is fascinating, the bar for demonstrating this is quite high. One convincing approach would be to observe these states (or similar ones) having evolved independently, concurrent with separate changes in pregnancy complexity along the syngnathid phylogeny. A single event (evolved pregnancy complexity in the *Hippocampus/Syngnathus* lineage), as studied in the current manuscript, doesn’t rule out that the evolution of these phenotypes isn’t just coincidental. For example, evolutionary loss of spleen could be associated with the evolution of another derived trait in syngnathids via developmental genetic pleiotropy, etc. An expanded comparative approach would make for a much more compelling test of concerted evolution.

Secondary comments:

1. Abstract, line 37: “secondarily” is not necessary. The traits would be simply “lost.”
2. Introduction, line 60: role of placenta in providing nutrients and oxygen needs citations.
3. Results, lines 97-98: Syngnathid reduction in genome size occurred prior to the evolution of seahorses, and separate from pouch complexity evolution (see Roth et al. 2020). Also, protein-coding regions make up a small fraction of total genome length. Loss of immune system genes is most likely not a significant contributor based on the available genome data.
4. Results, lines 102-105, and elsewhere in the MS: Many of the insights regarding reduction in immune gene repertoire in syngnathid lineages were previously published (Lin et al. 2016, Roth et al. 2020). The authors should clarify which insights are new from this study, versus confirmatory of previous studies, with relevant citations.
5. Results, lines 141-143: Concluding that a specific mutation in the past “led to” a derived trait is always difficult (in many cases impossible) to establish. An alternative is that the observed substitution occurred due to relaxed negative selection on the gene after an initial disruption (and trait loss) to a gene(s) elsewhere in the regulatory network.
6. Fig. 3 (C), and lines 633-634: In the supplement the authors should describe, at least briefly, how the spleen detection assays in zebrafish were performed. Were they randomized and done blind with respect to mutant status? Also, photographs of all 78 fish should be available in the supplement or in a repository, for readers to assess. If there is variation, or deviation from the “representative” individuals in the figure, it is important for readers to note.

7. Extended Data Fig. 10, line 876: This experiment (comparison of pregnant and non-pregnant seahorses) is not described anywhere in the MS, as far as I could tell. Are these data and figure taken from a previously published paper? If so proper citations, and an explanation, are in order.

8. Methods, lines 498-499: Why did the authors remove gene families with "multiple functional annotations" from the expansion/contraction analysis? This has the potential to result in systematic bias in the KEGG/GO enrichment analyses. A statistical approach that incorporates fractional functional weightings would be more appropriate, and retain potentially important gene families in the analysis.

9. What KEGG pathways or GO terms were enriched in the seahorse- (or syngnathid-) expanded gene families? If immune function categories are also overrepresented among expanded families, it implies that gene turnover in general, and not just loss, is at play, complicating the story somewhat.

10. Were the 336 positively selected genes identified with the false discovery rate controlled? If not, these adjustments should be made whenever many hypothesis tests are conducted together.

11. Methods, lines 517-526: The method of identifying "rapidly evolving" genes is fundamentally problematic. The authors claim that a branch-specific omega (dN/dS ratio) higher than background is used to identify rapidly evolving genes. Omega is a measure of selection. It is a rate ratio, not a rate. Indeed, slowly evolving proteins can have a high omega, when there are only a few DNA substitutions but they are all or almost all non-synonymous. If the goal is to identify rapidly evolving proteins, the appropriate metric would be a dN (non-synonymous substitution rate) that is higher than background.

12. Based on the supplementary information, many of the identified immune system gene losses and/or modifications are not unique to seahorses, but rather characteristic of syngnathids in general. While these changes could still be related to the evolution of male brooding in the syngnathid ancestor, they are not consistent with links to evolved, complex pregnancy in seahorses. Family-wide reduction in immune repertoire should be discussed. Also see above comment (#4) about gene and gene family losses previously published by the authors.

13. Hippocampus and Syngnathus lineages share a recent common ancestor relative to the other (basally located) syngnathid lineages examined, but there is still deep divergence between the two considering the entire phylogeny (see Stiller et al. 2022, BMC Biology). Many lineages (e.g. genera) share more recent common ancestry with seahorses and Syngnathus pipefishes, respectively, so the information missing from those parts of the tree should be emphasized and discussed somewhere. Those lineages exhibit variation in pregnancy complexity, so they are an important piece of the puzzle.

14. At least one "in-house" Perl and one R script are mentioned in the methods, which are not provided in the supplement or a code repository.

Point-by-point response to reviewer's comments

Reviewer #1 (Remarks to the Author):

In this study Liu et al. have compared two high quality genomes of seahorses (generated in this study) with other teleost genomes (previously published) focusing on immune-related genes. The seahorses and the related pipefishes are peculiar because they have evolved male-pregnancy and have undergone severe changes in their adaptive immune system. Furthermore, seahorses and pipefishes within the genus *Syngnathus* have developed closed (male) brooding organs and they also lost their entire spleen (asplenia). Liu and colleagues have found that a single amino acid substitution (Thr – Ala) in the transcription factor *tlx1* is associated with asplenia in seahorses. They further showed by generating CRISPR-Cas9 knockout and point mutation lines in zebrafish lead to loss of spleen in this teleost model system.

This is a well conducted and well-written study providing important comparative immunogenomic results that have been tested experimentally in zebrafish. The results contribute to our understanding of the peculiar phenomenon of male pregnancy, identification of a mutation that leads to loss of spleen as well as immunogenomic modifications in the *Syngnathidae*. I have a few questions for clarifications and some minor comments.

Response: Thank you for your complimentary remarks. We appreciate your insightful and helpful comments, which greatly helped to improve our manuscript in this revision. We have addressed all your specific comments and provided a point-by-point response below.

Questions for clarifications:

Comment 1: In the introduction I think it is appropriate to mention that there are two versions of adaptive immunity (MHC and VLR) here or specify that you are referring to the MHC/BCR/TCR system of adaptive immunity (line 45).

Response: Thank you for this suggestion. In the revised manuscript, we changed the sentence and specified that we are referring to the MHC/BCR/TCR system in the jawed vertebrate (Line 43-45).

Comment 2: Lines 62-63: Does *syngnathids* refer to both pipefishes and seahorses, or just pipefishes? Pipefishes have not been mentioned so far in the text - just seahorses. Maybe just use seahorses here to avoid confusion? – or explain

Response: Thank you for your comment. According to your suggestion, we changed “*syngnathids*” to “seahorses” in Line 64.

Comment 3: It is stated in lines 219-220 that seahorses and pipefishes have a functional loss of the MHC II pathway. However, in Fig 4 MHC II genes are shown to be present in seahorses (absent in pipefishes). It is referred to Roth et al. 2020 (PNAS). This is still

a bit confusing, particularly for a non-pipefish/seahorse scientist – and should be better explained.

Response: Thank you for pointing out this unclarity in our manuscript. Based on short-read genome data (Roth *et al.* 2020), we have found that all assessed representatives of *Syngnathus* have lost several genes in the MHC II pathway. In contrast, the representatives of the *Hippocampus* have only lost certain exons of *cd74*, a central gene in the MHC II pathway. Without these exons, the MHC II pathway was hypothesized to be unfunctional in seahorses in its original form and that genes were subject to neofunctionalization (Roth *et al.* 2020). This rearrangement of differing MHC II pathways between the genera *Hippocampus* and *Syngnathus* suggests a strong selection for reduction of “immunological vigilance” displayed by the MHC class II pathway, which illustrates the remarkable flexibility of the vertebrate immune system in general, as fish evidently somehow immunologically compensate for this loss. It was hypothesized that modification of adaptive immunity might be correlated to the degree of male pregnancy, which would explain these unexpected findings (Roth *et al.* 2020), and therefore, to avoid misunderstanding, we have removed “e.g. the functional loss of the MHC II pathway” and rephrased the sentence in the revised manuscript (Line 234-236).

Comment 4: In other “systems” where MHC II is lost such as in Lophiiformes (e.g. monkfish) and gadiformes (e.g Atlantic cod) both these species have spleen (and codfishes in general have spleen, but MHC II is lost from the entire lineage). I think that this issue should be touched upon in the Discussion because there is obviously not a clear connection between loss of spleen and loss of MHC II. Furthermore, according to this MS, it is not clear if the Syngnathides with an alanine in TLX1 actually possess a spleen or not. This issue may also be discussed.

Response: Thank you for your insightful comment. We understood that we did not provide a clear connection between loss of spleen and loss of MHC II in the previous version of the manuscript. We address this issue in more detail in the reply to reviewer 2’s comments, but, briefly, we show now the splenic phenotype for a number of syngnathid species whose *tlx1* sequence was analyzed. Most importantly, we dissected *Syngnathoides biaculeatus* (with an alanine in TLX1) individuals, and we found that they all possessed a spleen, which is consistent with our prediction that species with an alanine retained a spleen (Supplementary Figure 11c). We added the description and the anatomic results in the supplementary results. Also, according to your suggestion, we have cited below four references and discuss this issue now: “As a vital part of the adaptive immune system, the lack of the MHC class II pathway apparently does not always lead to asplenia, since spleens are present in other species that lost MHCII, as is known from Gadiformes (e.g., Atlantic cod) and the non-parasitizing anglerfish *Lophius piscatorius*” in the revised manuscript (Line 238-241).

References:

Star B, Nederbragt A J, Jentoft S, *et al.* The genome sequence of Atlantic cod reveals a unique immune system. *Nature*, 2011, 477(7363): 207-210.

Malmstrøm M, Matschiner M, Tørresen O K, *et al.* Evolution of the immune system influences speciation rates in teleost fishes. *Nature Genetics*, 2016, 48(10): 1204-1210.

Dubin A, Jørgensen T E, Moum T, *et al.* Complete loss of the MHC II pathway in an anglerfish, *Lophius piscatorius*. *Biology letters*, 2019, 15(10): 20190594.

Guslund N C, Krabberød A K, Nørstebø S F, *et al.* Lymphocyte subsets in Atlantic cod (*Gadus morhua*) interrogated by single-cell sequencing. *Communications Biology*, 2022, 5(1): 1-9.

Comment 5: Extended Fig 3b: the Figure shows that the T - A amino acid (point) mutation is not present in all Syngnathidae. Indeed, the figure shows *Oostethus manadensis* that display an L in the T/A site as well as T in the adjacent site. As far as I could see this is not commented on, and I am wondering what this could imply.

Response: Thanks for your comment. The family Syngnathidae is a large (>350 species) and diverse clade of morphologically unique teleosts. They can be divide into two subfamilies: the *Nerophinae* and the *Syngnathinae*. Using morphological, histological and Micro CT methods, we provided the splenic phenotype for a number of syngnathid species now. In detail, we found the genera *Hippocampus* and *Syngnathus* (both belonging to *Syngnathinae*) do not possess an unambiguous spleen (Supplementary Figure 11a-b). In addition, due to the approach of sample limitations, we cannot examine the *Oostethus manadensis*, but we checked the splenic phenotype of *Syngnathoides biaculeatus* (belongs to the subfamily *Syngnathinae*) and *Nerophis ophidion* (belongs to the subfamily *Nerophinae*). Not surprisingly, *Syngnathoides biaculeatus* and *Nerophis ophidion* possessed a spleen, which is consistent with our results that species with an alanine retained/or not threonine retained a spleen (Supplementary Figure 11c-d). Taken together, the TLX1 sequences exhibit a number of different amino acid substitutions (A to T or A to L) in these species, but based on the multiple methods, we observed that the loss of the typical spleen only occurs in the species with the “A to T” amino acid in the TXL1 gene, while other amino acid substitutions do not seem to affect spleen development. We point this out more clearly in our revised manuscript now in Line 133-141 “As for the subfamily *Nerophinae*, we found the TLX1 sequences exhibit amino acid substitutions of A to L and A to I in *Oostethus manadensis* and *Nerophis ophidion*, respectively (Extended Data Fig. 3b). We also provided the splenic phenotype for a number of syngnathid species using morphological, histological, and Micro CT methods. Dissections showed that species of the genera *Hippocampus* and *Syngnathus* (both belonging to *Syngnathinae*) have evolutionarily lost an unambiguous spleen, but not *Syngnathoides biaculeatus* (belongs to the subfamily *Syngnathinae*) nor *Nerophis ophidion* (that belong to the subfamily *Nerophinae*) (Supplementary Figure 11)”.

Minor comments:

Comment 6: Abstract (lines 35-36) it is possible to read this as if the CRISPR-Cas9 experiments were conducted on seahorses and not zebrafish.

Response: Thank you for pointing out this ambiguity. We revised this sentence and made the intended meaning more clear now (Line 35-36).

Comment 7: Line 131: alanine

Response: Thank you for pointing out the spelling error. We changed “Alanin” to “Alanine” in the revised manuscript (Line 133).

Comment 8: Line 162: add a short sentence here about what the expression profile is like in wildtype vs asplenia zebrafish? As of now, the sentence does not provide much insight.

Response: Thank you for your insightful suggestion. We have rewritten the sentence and added more information on the expression profile change of these genes in the revised manuscript (Lines 173-176).

Comment 9: The interesting study by Parker et al. (2022) on single cell transcriptomics of immune cells in pipefish is not mentioned in the Results/Discussion. The results here are relevant to be discussed.

Response: Thank you very much for your comment, we’ve changed this in the revised version of the manuscript (Line 264-267).

Comment 10: Figure 3: I understand that the knock out/point mutation lines were sequenced by the old Sanger method (perfectly suited for this!) – but it is to this reviewer a bit confusing since sequencing otherwise is done by high throughput methods. Just explain that this is Sanger sequencing in the Fig legend and in Methods.

Response: Thank you for pointing out this omission in the figure legend. In the revised manuscript, we’ve added and specified the sequencing method of the CRISPR/Cas9 editing zebrafish lines in the figure legend and the methods section (Lines 462-463 and Lines 722-723).

Reviewer #2 (Remarks to the Author):

Summary:

In this manuscript Liu et al. report on two overarching observations: 1. Loss of suites of genes related to immune system development and function in the seahorse - and with some generality the syngnathid - lineage, and 2. A derived (possibly synapomorphic with respect to a clade that includes seahorses) amino acid substitution in the transcription factor *tlx1*, which is known to influence development of the spleen from previous gene editing studies in mouse and zebrafish. The impetus is that these two patterns may be related (have “coevolved” in the authors’ words), in the context of male pregnancy, although this is the type of cause-and-effect connection that is difficult to establish from observed states in extant organisms, especially with just a single lineage (and evolutionary event), as considered in this study.

To confirm the loss of immune system genes, which some of the authors have reported previously in other papers, the researchers sequenced two additional seahorse genomes and conducted comparative gene family evolution analyses using seahorse genomic resources, several additional syngnathid genomes, and a sample of more distantly related teleost (plus gar) genomes. These genomic resources were also used to identify species with a derived *tlx1* substitution, which include seahorses and members of the *Syngnathus* pipefish genus. The authors then used CRISPR-Cas9 genome editing to knockout *tlx1* in zebrafish, and they also introduced the Hippocampus/*Syngnathus* substitution as a point mutation in a different zebrafish line. According to the authors, both mutations resulted in anatomical asplenia with full penetrance, supporting a role for the gene, and more precisely the specific point mutation, in zebrafish asplenia. They also compare gene expression profiles for various tissues between the two mutants and conclude that the knockout mutant exhibits more transcriptional disruption (more DE genes relative to WT) than the point mutant. The authors report that asplenia is observed in seahorses and *Syngnathus* pipefishes, which suggests the point mutation in zebrafish induces phenocopy, but there are no methodological details, histology, labeling assays, or imaging to describe the syngnathid asplenia phenotype, and the validity of the single supporting reference is questionable (see below).

In general, I view the zebrafish gene editing as main strength of the paper, with the potential to inform on more specific mechanisms by which *tlx1* interacts with other genes to affect spleen development in vertebrates. The primary shortcoming (see below for details) is the lack of convincing support for asplenia in syngnathids, and for direct, systematic relationships between the reported phenotype, immune system gene repertoire evolution, and male pregnancy. For these reasons I think the paper could be a very valuable contribution to the *tlx1* and spleen development literature, published in the appropriate developmental journal, but I cannot recommend publication in Nature Communications.

Response: Thank you for your comments and input on our manuscript. We appreciate

your insightful and helpful comments, which greatly helped to improve our manuscript in this revision. Specifically, we added a more thorough analysis of the splenic phenotype using multiple approaches (morphology, histology, Micro CT scanning and transcriptomic analysis) in a number of syngnathids. Meanwhile, we also added an expanded comparative approach of character's state reconstruction of asplenia, immune gene simplification and male pregnancy. We have addressed all your specific comments and provided a point-by-point response below.

Primary comments:

Comment 1: An especially significant concern is the lack of validation of asplenia in *Hippocampus*, *Syngnathus*, and other syngnathids. Asplenia is purported by the authors, based on a single reference (Matsunaga and Rahman 1998), which was also the source for the other, secondary citation (Roth et al. 2020). Unfortunately, Matsunaga and Rahman report only that “seahorses might have lost GALT,” based on their inability to identify a spleen from the dissected GI tract of a single, 10 cm *Hippocampus kuda* specimen. No experimental validation is reported in the current manuscript. Because this phenotype (anatomical asplenia) is so essential to the authors' thesis, a much more rigorous approach, replicated in multiple individuals from multiple syngnathid species, is crucial for establishment as fact. Expectations for this kind of stated finding should probably include more than one assay: options are histology, in situ hybridization with a spleen-diagnostic probe, tomographic imaging, etc.

Response: Thank you for your insightful comments. We agree that more data should have been provided in the first version of the manuscript to support the hypothesis that adult *Syngnathus* and *Hippocampus* are in fact asplenic. According to your suggestion, we have now dissected several syngnathid species, including such predicted by our work to have a spleen (*Syngnathoides biaculeatus* and *Nerophis ophidion*) and such to feature asplenia (*Syngnathus typhle*, *Hippocampus erectus* and *Hippocampus abdominalis*), and provide evidence for their splenic phenotype. Specifically, we provide anatomical photos, Micro CT scans and spleen transcriptome of the *Syngnathoides biaculeatus* where a spleen could unambiguously be identified, which was also confirmed in *Nerophis ophidion*. In addition, we also support our findings in *Hippocampus* and *Syngnathus* by providing anatomical asplenia phenotype. Our results showed that those predicted to have lost the spleen (*Syngnathus* & *Hippocampus*) featured asplenia (no unambiguous spleen). We have identified a small white organ in both species, however, comparative transcriptomics suggest distinct expression patterns in this small white organ in contrast to the spleen found in another syngnathid. This implies that this small white organ cannot be a functional spleen, and we thus termed representatives of *Syngnathus* and *Hippocampus* having a functional “asplenia” (but it might be a vestigial organ indeed) (as shown in the figure below; Supplementary Figure 11, Supplementary Table 27). *In situ* hybridization of *tlx1* also revealed the asplenia of the embryo in *Hippocampus erectus* in our study (Extended Data Fig. 5a). Not surprisingly, *Syngnathoides biaculeatus* possessed a spleen, which is consistent with our results that species with an alanine retained a spleen (c-d, red arrow), which we now

describe in the manuscript Line 133-141. The detailed methods were added in Line 654-672. Altogether, in the revised version of our manuscript there is now stronger support for our hypothesis that the identified point mutation is likely causally linked to the asplenia in syngnathids. Unfortunately, we could not dissect any seadragon specimens, as we only had access to four individuals, and they were all sacrificed for our previous genomic and transcriptomic analysis work.

Comment 2: Particularly important is confirmation of asplenia (or lack thereof) also in the other (non-*Hippocampus/Syngnathus*) syngnathid species for which *tlx1* was examined. For example, if asplenia is confirmed in all syngnathids, then the position 208 A-T substitution can't fully explain asplenia, because *Syngnathoides*, *Phyllopteryx*, and *Oostethus* all have the ancestral or different derived (*Oostethus*) states. The authors need to convincingly establish asplenia status for all or most of the above lineages to support their conclusions. Because this author group appears to have published on these taxa before, they should have access to the required samples.

Response: Thank you very much for your comments. We agree with the reviewer that the confirmation of asplenia in other (non-*Hippocampus/Syngnathus*) representatives of syngnathids is particularly important, thus, we have now added the data in the revised manuscript. The family Syngnathidae is a large (>350 species) and diverse clade of morphologically unique teleosts. They are divided into two subfamilies: the *Nerophinae* and the *Syngnathinae*. Due to the impact of the global epidemic, the availability of ornamental fishes is limited and not for all species of interest individuals could be obtained, but we have examined the splenic phenotype for *Syngnathoides biaculeatus* (belongs to the subfamily *Syngnathinae*) and *Nerophis ophidion* (belongs to the

subfamily *Nerophinae*). Our results showed that *Syngnathoides biaculeatus* and *Nerophis ophidion* possessed a spleen, which is consistent with our predictions that species with an alanine retained/or not threonine retained a spleen (Supplementary Figure 11). This result is not surprising that within ~350 syngnathids species that have different brood pouch types and variable body characteristics we can observe some diversity in this amino acid locus. Our experiments in zebrafish demonstrated that the replacement of 208A-T can cause asplenia in zebrafish, while the validation of its adjacent site (207A-T) retained the spleen throughout the ontogeny, suggesting that this amino acid replacement is not a coincidence. Taken together, the TLX1 sequences exhibit a number of different amino acid substitution (T to A or L to A) in these species, but based on the multiple methods, we observe that the loss of the typical spleen only occurs in the species with the “A to T” amino acid in the TXL1 gene, while other amino acid substitutions do not seem to affect spleen development. In conclusion, to be more rigorous in our results, we have emphasized in the revised manuscript that alterations in A-T may be a fundamental cause of spleen loss in *Hippocampus* and *Syngnathus*. As more samples are obtained and more genomes become available, we will conduct more in-depth studies on the problem of spleen loss in other species in future work.

Comment 3: While the premise of concerted evolution of sweeping loss of immune system genes, loss of the spleen, and complex brooding tissues is fascinating, the bar for demonstrating this is quite high. One convincing approach would be to observe these states (or similar ones) having evolved independently, concurrent with separate changes in pregnancy complexity along the syngnathid phylogeny. A single event (evolved pregnancy complexity in the *Hippocampus*/*Syngnathus* lineage), as studied in the current manuscript, doesn't rule out that the evolution of these phenotypes isn't just coincidental. For example, evolutionary loss of spleen could be associated with the evolution of another derived trait in syngnathids via developmental genetic pleiotropy, etc. An expanded comparative approach would make for a much more compelling test of concerted evolution.

Response: Thank you for the comment. We agree with the reviewer that the expanded comparative approach would make for a much more compelling test of concerted evolution. Therefore, in the revised manuscript, we added more syngnathid species including more seahorses, pipefishes and seadragons, and conducted characters state reconstruction, including the amino acid replacement of *tlx1*, splenic phenotype, male pregnancy and immune response simplification (Line 282-291, Line 800-811). As shown in the figure below, our result revealed that the A to T mutation only occurred in *Hippocampus* and *Syngnathus*, which possess asplenia (no unambiguous spleen). In addition, *Hippocampus* and *Syngnathus* exhibited completely closed brooding pouches and inverted pouch folds, respectively, which is a relatively complex type of brood pouch and male pregnancy along the syngnathid phylogeny. Our results also showed that some immune-related genes (including the lost/contracted genes we identified in this study like *batf3*, *C4*, *C3*, *CD5*, and *foxp3*) were also lost in Syngnathidae species including *Nerophis ophidion*, *Oosththus manadensis*, *Syngnathoides biaculeatus*, *Phyllopteryx taeniolatus*, *Syngnathus scovelli*, *Syngnathus typhle*, *Syngnathus acus*,

Hippocampus comes, *Hippocampus erectus*, *Hippocampus zosterae* and *Hippocampus mohnikei*. Indeed, as a conserved organ, the loss of the spleen may not only be accompanied by the evolution of male pregnancy but may also bring about changes in some other traits, such as the observed immune response changes. Our results revealed that asplenia and male pregnancy characters evolved independently but cooccurred on the same branches (*Hippocampus* and *Syngnathus*) as a form of concerted evolution at the same time (Extended Data Fig. 10). Hence, our speculation that the synchronized evolution of asplenia and male pregnancy may not be accidental but instead an evolutionary link exists. However, as we do not have more evidence to support this speculation, we weaken our claims about the co-evolution in the revised article and use the more appropriate cooccurred in order to make the article more rigorous.

Secondary comments:

Comment 4: Abstract, line 37: “secondarily” is not necessary. The traits would be simply “lost.”

Response: Thank you for your comment. We deleted “secondarily” in the revised manuscript (Line 37).

Comment 5: Introduction, line 60: role of placenta in providing nutrients and oxygen needs citations.

Response: Thank you for pointing out this error. Now, we cite the following references in the revised manuscript (Line 62).

References:

Skalkos Z M G, Van Dyke J U, Whittington C M. Paternal nutrient provisioning during male pregnancy in the seahorse *Hippocampus abdominalis*. *Journal of Comparative Physiology B*, 2020, 190(5): 547-556.

Dudley J S, Hannaford P, Dowland S N, *et al.* Structural changes to the brood pouch of male pregnant seahorses (*Hippocampus abdominalis*) facilitate exchange between father and embryos. *Placenta*, 2021, 114: 115-123.

Stölting K N, Wilson A B. Male pregnancy in seahorses and pipefish: beyond the mammalian model. *BioEssays*, 2007, 29(9): 884-896.

Comment 6: Results, lines 97-98: Syngnathid reduction in genome size occurred prior to the evolution of seahorses, and separate from pouch complexity evolution (see Roth *et al.* 2020). Also, protein-coding regions make up a small fraction of total genome length. Loss of immune system genes is most likely not a significant contributor based on the available genome data.

Response: Thank you for your comment. Based on the available genome data, we agree with the reviewer that the loss of immune system genes is likely not a significant contributor to the reduction in Syngnathid genome size. Therefore, we remove the sentence “which can partly explain their relatively small genome sizes” in the revised manuscript.

Comment 7: Results, lines 102-105, and elsewhere in the MS: Many of the insights regarding reduction in immune gene repertoire in syngnathid lineages were previously published (Lin *et al.* 2016, Roth *et al.* 2020). The authors should clarify which insights are new from this study, versus confirmatory of previous studies, with relevant citations.

Response: Thank you for this suggestion. Our previous studies in Lin *et al.* (*Nature*, 2016) revealed the contraction of immune genes (such as C-type lectins, Macrophage mannose receptor and protein NLRC3) only in *Hippocampus comes*. Meanwhile, another paper we published in PNAS (Roth *et al.* 2020) provides further insights into canonical vertebrate immunity in seahorses and pipefishes (mainly focusing on the MHC pathway). With high-quality genomes sequenced by PacBio and Nanopore technologies, our present study not only confirms previous studies but also provides new insights into immune gene reductions (including the *batf3*, *cd5*, *foxp3*, *c4*, *etc.*). Therefore, according to the reviewer’s suggestion, we have added the citations and emphasized our new findings in the revised manuscript to better state which novel findings we report in this study.

Comment 8: Results, lines 141-143: Concluding that a specific mutation in the past “led to” a derived trait is always difficult (in many cases impossible) to establish. An

alternative is that the observed substitution occurred due to relaxed negative selection on the gene after an initial disruption (and trait loss) to a gene(s) elsewhere in the regulatory network.

Response: Thank you for the insightful suggestion. We rephrased the sentence to: "...suggesting that the seahorse-specific mutation is causally linked to a functional alteration of *tlx1*, which leads to functional asplenia. Whether this mutation originally caused functional asplenia in this syngnathid lineage, or whether another mutation was causal, removed stabilizing selection from genes involved in spleen development, and *tlx1* mutated only then, cannot be resolved using our data-set..." in Line 150-155.

Comment 9: Fig. 3 (C), and lines 633-634: In the supplement the authors should describe, at least briefly, how the spleen detection assays in zebrafish were performed. Were they randomized and done blind with respect to mutant status? Also, photographs of all 78 fish should be available in the supplement or in a repository, for readers to assess. If there is variation, or deviation from the "representative" individuals in the figure, it is important for readers to note.

Response: Thank you for your comment. The spleen detection assays in zebrafish were performed according to the reference by Xie et al. Briefly, after euthanasia with 200 ng/ml MS-222 (Sigma-Aldrich, USA), the whole-mounts of abdominal organs were carefully removed from wild-type (23 fish) zebrafish as well as *tlx1*^Δ (11 fish), *tlx1*^{A208T} (27 fish), and *tlx1*^{A207T} (17 fish) mutants (Fig. 3). The organs were then fixed in 4% PFA for 5 min, and images were obtained using a MVX10 camera (Olympus, Japan). Yes, all zebrafishes checked were sampled randomized and done blind with respect to mutant status. We describe the method in detail in the revised manuscript (Line 728-733). In addition, photographs of other 74 fish were added in the supplementary materials (Supplementary Figure 12-14). We also attached the photographs below.

Reference:

Lang Xie, Yixi Tao, Ronghua Wu, *et al.* Congenital asplenia due to a *tlx1* mutation reduces resistance to *Aeromonas hydrophila* infection in zebrafish. *Fish and Shellfish Immunology*, 2019, 95: 538-545.

wild type (22 individuals)

tlx1^{A208T} (26 individuals)

tlx1^Δ (10 individuals)

tlx1^{A207T} (16 individuals)

Comment 10: Extended Data Fig. 10, line 876: This experiment (comparison of pregnant and non-pregnant seahorses) is not described anywhere in the MS, as far as I could tell. Are these data and figure taken from a previously published paper? If so proper citations, and an explanation, are in order.

Response: Thank you for pointing out the error, and we are sorry for the negligence of the detailed method description. The comparison of pregnant and non-pregnant seahorses' brood pouches was conducted in the present study using Real-time qPCR. The primers of these detected genes are listed in Supplementary Table 32. According to the reviewer's suggestion, we have added the detailed methods of this experiment in the revised manuscript (Line 812-820).

Comment 11: Methods, lines 498-499: Why did the authors remove gene families with "multiple functional annotations" from the expansion/contraction analysis? This has the potential to result in systematic bias in the KEGG/GO enrichment analyses. A statistical approach that incorporates fractional functional weightings would be more appropriate and retain potentially important gene families in the analysis.

Response: Thank you for this suggestion. We removed the "multiple functional annotations" gene families from the expansion/contraction analysis to avoid errors caused by sequence differences. We agree with the reviewer that a statistical approach that incorporates fractional functional weightings would be more appropriate and retain potentially important gene families in the analysis. We redid the analysis based on your suggestion using fractional functional weightings and updated the methods and results in the KEGG/GO enrichment analyses. Our new KEGG enrichment of contracted gene families in seahorse species also revealed the signatures related to immune response pathways, such as the allograft rejection and antigen processing and presentation, which is similar to our previous results. In addition, we also revised the enrichment analysis of the gene losses, positive selected genes, rapidly evolving genes and lineage-specific mutated genes in the revised manuscript (Line 562-569; Figure 1c; Extended Data Fig. 1b; Supplementary Figure 4-5; Supplementary Table 14-16, 18-22, 24, 25, highlighted in yellow).

Comment 12: What KEGG pathways or GO terms were enriched in the seahorse- (or syngnathid-) expanded gene families? If immune function categories are also overrepresented among expanded families, it implies that gene turnover in general, and not just loss, is at play, complicating the story somewhat.

Response: Thank you for your insightful suggestion. Accordingly, we have reanalyzed the GO terms of the expanded gene families and the contracted gene families in the seahorses in the revised manuscript (Supplementary Table 14, highlighted in yellow). Our results showed that the contracted gene families were enriched in the GO terms related to immune function categories, such as inflammasome complex, MHC protein complex, chemokine activity, immune response, etc. However, we did not find the GO terms of the expanded gene families related to immune function categories under the same constraints (with $Q\text{-value} < 0.05$).

Comment 13: Were the 336 positively selected genes identified with the false discovery rate controlled? If not, these adjustments should be made whenever many hypothesis tests are conducted together.

Response: We agree with the reviewers' comment and the 336 positively selected genes were identified with the controlled false discovery rate (FDR<0.05). To clarify this, we added the FDR value in the revised supplementary materials (Supplementary Table 17, highlighted in yellow) and added the mentioning the use of the false discovery rate-controlled method in Line 587-590.

Comment 14: Methods, lines 517-526: The method of identifying "rapidly evolving" genes is fundamentally problematic. The authors claim that a branch-specific omega (dN/dS ratio) higher than background is used to identify rapidly evolving genes. Omega is a measure of selection. It is a rate ratio, not a rate. Indeed, slowly evolving proteins can have a high omega, when there are only a few DNA substitutions but they are all or almost all non-synonymous. If the goal is to identify rapidly evolving proteins, the appropriate metric would be a dN (non-synonymous substitution rate) that is higher than background.

Response: Thank you for this suggestion. We first identified the rapidly evolving genes with a dN/dS ratio higher than the background based on the reference by Lin et al (Lin et al., 2019, *Science*). After careful consideration, we agree now with the reviewer's opinion that the dN value is an appropriate metric when identifying rapidly evolving genes. Thus, according to the reviewer's suggestion, we redid the analysis and updated the rapidly evolving genes in the revised manuscript (Supplementary Table 20).

Comment 15: Based on the supplementary information, many of the identified immune system gene losses and/or modifications are not unique to seahorses, but rather characteristic of syngnathids in general. While these changes could still be related to the evolution of male brooding in the syngnathid ancestor, they are not consistent with links to evolved, complex pregnancy in seahorses. Family-wide reduction in immune repertoire should be discussed. Also see above comment (#4) about gene and gene family losses previously published by the authors.

Response: Thank you for this suggestion. To confirm this, we have examined the presence and absence status of the immune-related genes (*batf3*, *foxp3*, *C3*, *C4*) that are missing in syngnathids in relatively closely related other genomes (*Aeoliscus strigatus* and *Fistularia tabacaria*). We found these four genes to be intact in both species (the sequences were provided in Supplementary Table 33). We agree with the reviewer's opinion that many of the identified immune system gene losses and/or modifications are general characteristics of syngnathids and added the sentences "In addition, as demonstrated here, this reduction in immune repertoire is shared across the family *Syngnathidae* and is not unique to seahorses; thus, further analyses will be necessary to evaluate the immune response strategy in species with different levels of pregnancy complexity in the future" in the revised manuscript (Line 287-291).

Comment 16: *Hippocampus* and *Syngnathus* lineages share a recent common ancestor

relative to the other (basally located) syngnathid lineages examined, but there is still deep divergence between the two considering the entire phylogeny (see Stiller et al. 2022, BMC Biology). Many lineages (e.g. genera) share more recent common ancestry with seahorses and *Syngnathus* pipefishes, respectively, so the information missing from those parts of the tree should be emphasized and discussed somewhere. Those lineages exhibit variation in pregnancy complexity, so they are an important piece of the puzzle.

Response: Thank you for the insightful suggestion. We agree with the reviewer that there is still deep divergence between the *Hippocampus* and *Syngnathus* lineages considering the entire phylogeny, and information on those lineages in between *Syngnathus* and *Hippocampus* would be important to derive more informed conclusions. However, the genomes of these intermediate species are not available yet, preventing us to perform such analyses. So, according to the reviewer's suggestion, we cited the reference (Stiller *et al.* 2022) and added the sentence "Considering the entire syngnathid phylogeny, there is still deep divergence between the *Hippocampus* and *Syngnathus* lineages, and the splenic phenotype and *tlx1* gene sequence information of lineages more closely related to *Hippocampus*, which are missing in our study, would improve the resolution of the dataset and thus allow to formulate better informed conclusions. Therefore, we encourage further studies filling these gaps in genomic knowledge." in the revised manuscript (Line 306-312).

Comment 17: At least one "in-house" Perl and one R script are mentioned in the methods, which are not provided in the supplement or a code repository.

Response: Thanks for pointing out this omission. We added the codes of three "in-house" Perl (Line 545, 547, and 760) and one R script (Line 576) in the Supplementary Code Repository of the revised manuscript.

Reviewers' Comments:

Reviewer #1:

Remarks to the Author:

The authors have done a very good job in revising the ms. All my main points have been adequately addressed. I am also happy to see that the authors have included both new extended Figures and Supplementary Figures addressing some of my main points, as well as some of the others reviewers' points (we had "overlapping" points re. asplenia). I think this work is will be a very valuable contribution to Nature Communications.

Reviewer #2:

Remarks to the Author:

The authors have made substantial revisions to address the original, primary concerns of both reviewers, and I thank them for their efforts. I agree with the authors' decision to temper language in the latest version of the manuscript, particularly related to issues of potential causality and concerted evolution. I also appreciate the additional clarifications and analyses regarding the broader phylogenetic context of the questions at hand.

That said, I recommend two minor changes.

First, given the authors' response letter, and language adjustment choices throughout the manuscript, it seems like the title of the paper should be adjusted commensurately. Perhaps something like "Immunogenetic losses cooccurred with seahorse male pregnancy and mutation in *tlx1* accompanied functional asplenia," would better match the current version of the text.

Second, I originally suggested controlling the false discovery rate (FDR) specifically for the positively selected genes analysis, and generally whenever many parallel hypothesis tests are conducted simultaneously. It looks like the authors did include FDR-adjusted p-values for the positive selection analysis (Supp. Table 17), but those were apparently calculated using only the list of p-values < 0.05. To properly control the FDR, p-values from all original hypothesis tests must be included in the calculations. This issue should be corrected in the case of this analysis, and for other analyses throughout the paper where relevant.

Point-by-point response to reviewer's comments

Reviewer #1 (Remarks to the Author):

The authors have done a very good job in revising the ms. All my main points have been adequately addressed. I am also happy to see that the authors have included both new extended Figures and Supplementary Figures addressing some of my main points, as well as some of the others reviewers' points (we had "overlapping" points re. asplenia). I think this work will be a very valuable contribution to *Nature Communications*.

Response: Thank you for reading the manuscript and the complimentary remarks. We appreciate the reviewer's insightful and helpful comments, which greatly helped to improve our manuscript during the revision.

Reviewer #2 (Remarks to the Author):

The authors have made substantial revisions to address the original, primary concerns of both reviewers, and I thank them for their efforts. I agree with the authors' decision to temper language in the latest version of the manuscript, particularly related to issues of potential causality and concerted evolution. I also appreciate the additional clarifications and analyses regarding the broader phylogenetic context of the questions at hand.

That said, I recommend two minor changes.

Response: We appreciate the reviewer's careful reading of the manuscript and constructive remarks. Your insightful and helpful comments have greatly helped to improve our manuscript during the revision. We have addressed all your specific comments and provided a point-by-point response below.

Comment 1: First, given the authors' response letter, and language adjustment choices throughout the manuscript, it seems like the title of the paper should be adjusted commensurately. Perhaps something like "Immunogenetic losses cooccurred with seahorse male pregnancy and mutation in *tlx1* accompanied functional asplenia," would better match the current version of the text.

Response: Thank you for your comment. We agreed with the reviewer's suggestion for the title. We have changed the title to "Immunogenetic losses co-occurred with seahorse male pregnancy and mutation in *tlx1* accompanied functional asplenia" in the revised manuscript (Lines 1-2).

Comment 2: Second, I originally suggested controlling the false discovery rate (FDR) specifically for the positively selected genes analysis, and generally whenever many parallel hypothesis tests are conducted simultaneously. It looks like the authors did include FDR-adjusted p-values for the positive selection analysis (Supp. Table 17), but those were apparently calculated using only the list of p-values < 0.05. To

properly control the FDR, p-values from all original hypothesis tests must be included in the calculations. This issue should be corrected in the case of this analysis, and for other analyses throughout the paper where relevant.

Response: We appreciate the reviewer's suggestion. Thank you for pointing out the omission in the FDR-adjusted p-values tests. According to the review's suggestion, we've added the list of p-values and FDR-adjusted p-values for the analysis of the positively selected genes and the rapidly evolving genes in the revised manuscript using p-values from all original hypothesis tests (Supplementary Data 4-5).